# Generalized Logit Adjustment: Calibrating Fine-tuned Models by Removing Label Bias in Foundation Models

**Beier Zhu[1]**    **Kaihua Tang[1]**    **Qianru Sun[2]**    **Hanwang Zhang[1]**

[1]Nanyang Technological University    [2]Singapore Management University

`beier002@e.ntu.edu.sg, hanwangzhang@ntu.edu.sg`

## Abstract

Foundation models like CLIP allow zero-shot transfer on various tasks without additional training data. Yet, the zero-shot performance is less competitive than a fully supervised one. Thus, to enhance the performance, fine-tuning and ensembling are also commonly adopted to better fit the downstream tasks. However, we argue that such prior work has overlooked the inherent biases in foundation models. Due to the highly imbalanced Web-scale training set, these foundation models are inevitably skewed toward frequent semantics, and thus the subsequent fine-tuning or ensembling is still biased. In this study, we systematically examine the biases in foundation models and demonstrate the efficacy of our proposed Generalized Logit Adjustment (GLA) method. Note that bias estimation in foundation models is challenging, as most pre-train data cannot be explicitly accessed like in traditional long-tailed classification tasks. To this end, GLA has an optimization-based bias estimation approach for debiasing foundation models. As our work resolves a fundamental flaw in the pre-training, the proposed GLA demonstrates significant improvements across a diverse range of tasks: it achieves 1.5 pp accuracy gains on ImageNet, a large average improvement (1.4-4.6 pp) on 11 few-shot datasets, 2.4 pp gains on long-tailed classification. Codes are in `https://github.com/BeierZhu/GLA`.

## 1 Introduction

Thanks to the Web-scale data and self-supervised strategies, foundation models like CLIP [40] empower zero-shot transfer to a wide variety of domains [7, 1, 52]. However, the zero-shot performance is still weak on several domain-specific tasks such as differentiating models of cars, species of flowers, and variants of aircraft [40, 7]. Therefore, it is a common practice to improve the downstream performance via supervised fine-tuning on labeled data, *e.g.*, linear probing, prompt tuning [54, 55], and end-to-end fine-tuning.

However, fine-tuned models are easily biased: they are adept in exploiting spurious correlations that only hold on the downstream distribution [40, 39, 48, 55]. To improve the robustness, several studies [48, 55, 56] propose to combine fine-tuned models with zero-shot models. For example, WiSE-FT [48] ensembles the fine-tuned and zero-shot models in weight space and ProGrad [55] uses zero-shot predictions to regularize the fine-tuning gradient. The underlying assumption lies in that the zero-shot models are robust to distribution shifts [40], and their predictions are complementary to those of fine-tuned models [48].

Despite these methods exhibiting performance gains on both in-distribution and out-of-distribution evaluations, they all overlook the inherent bias originating from the foundation models. Specifically, the Web-scale data for pre-training foundation models exhibit a highly skewed distribution due to Zipf's law of nature [42]. The resulting foundation models develop a biased decision boundary that leads to a poor zero-shot performance on rare classes. As evidenced in Figure 1(a) and (b), the purple

37th Conference on Neural Information Processing Systems (NeurIPS 2023).

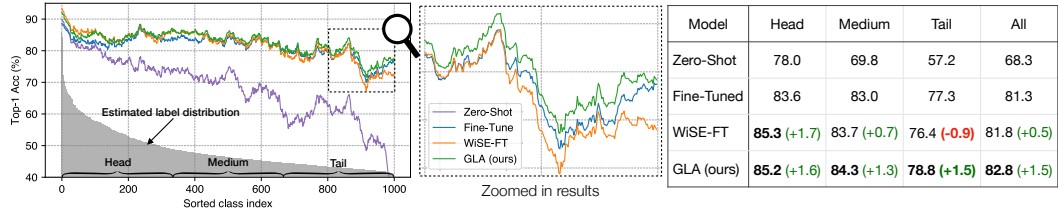

(a) Per class accuracy of different models on ImageNet      (b) Beak-down performance of different models on ImageNet

Figure 1: (a) Per class accuracy of CLIP-ViT/B16 on ImageNet. Class index are sorted using the estimated pre-training label prior. Curves are smoothed for better visualization. (b) Beak-down performance of different models on ImageNet. We equally divide the ImageNet classes into three subgroups, according to the class index. Existing ensemble methods like WiSE-FT [48] exhibits a clear performance loss on tail classes, while our GLA stands out for all three subgroups.

line encounters a dramatic drop, and the zero-shot performance of tail classes is significantly lower than that of head classes (57.2% *vs*. 78.0%). Existing ensemble methods like WiSE-FT [48] overlook the label bias, resulting in an improvement in top-1 (+0.5%) and head accuracy (+1.7%) while a noticeable degradation on the tail performances (−0.9%) in Figure 1(b). Another evidence is that the orange line (WiSE-FT) is below the blue line (fine-tuned models) for rare classes in Figure 1(a).

We propose Generalized Logit Adjustment (GLA), a simple post-hoc method consisting of two steps: **1)** removing the label bias of zero-shot model via estimating the label distribution in the pre-training dataset; **2)** ensembling the fine-tuned and debiased zero-shot models. As illustrated in Figure 1 (b), our GLA achieves consistent improvement across all three subgroups, particularly showing a significant gain on tail classes (+1.5%). Despite its simplicity, our GLA has a firm statistical grounding: it is the Bayes optimal classifier given the fine-tuned and zero-shot models, thus consistent for minimizing the error on a class-balanced target distribution (Section 4.2). It is worth noting that removing the bias of foundation models is challenging since the label distribution is often inaccessible due to privacy or copyright concerns. In this work, we only use the downstream labeled data and the zero-shot model to estimate the foundation label bias. Specifically, we formulate the problem by adjusting the margin of the zero-shot models such that the lowest error is achieved on the downstream dataset. This grounding translates into strong empirical performance on real-world datasets, covering few-shot, many-shot, and long-tail learning (Section 5).

The contributions and novelties of this work are summarized as follows:

- We point out the overlooked label bias in foundation models, which originates from the skewness of pre-training distribution that affects the performance of downstream tasks.

- We formalize the estimation of the label bias as a constrained optimization problem (Section 3.2) with theoretical justification (Section 4.3). The entire process does not require access to the pre-training dataset, making it practical for fine-tuning scenarios. We present Generalized Logit Adjustment (GLA) method, which ensembles the debiased zero-shot and fine-tuned models, and demonstrate its superiority over conventional fine-tuning and ensembling by proving it's a Bayes optimal classifier (Section 4.2).

- We build a comprehensive benchmark for evaluation, which considers three real-world settings and three fine-tuning paradigms. The settings are: 1) many-shot learning with abundant data 2) few-shot learning; and 3) long-tail classification, representing a more challenging scenario that combines many-shot and few-shot data (Section 5). The three fine-tuning paradigms include: 1) end-to-end fine-tuning; 2) linear probing; and 3) prompt tuning (Section 3.1).

- We demonstrate the efficacy of our proposed method GLA by conducting extensive experiments across various settings and fine-tuning paradigms. We observe 1 to 1.5 pp accuracy gains on ImageNet, large averaged improvement (1.4 to 4.6 pp) on 11 few-shot datasets and 2.4 pp averaged accuracy gains on long-tail datasets. (Section 5).

## 2 Related Work

**Image-text foundation models.** Foundation models pre-trained by contrastive objectives have set impressive milestones for image and text representation learning, with CLIP [40], ALIGN [23], CoCa [52] and Flamingo [1] being the exemplars. Such models exhibit impressive prompt-based zero-shot performance on various image recognition downstream tasks. Our method aims to reduce foundation model biases to boost performance in downstream tasks. While [2] also addresses word frequency bias, we differ in two key areas: Firstly, we debias zero-shot models using fixed prompts, whereas [2] refines the prompting process. Secondly, our GLA doesn't require access to a subset of the pre-training data.

**Ensembles.** Ensemble methods aim to boost performances by combining multiple networks, which can be either implemented by aggregating model outputs [12, 4, 28, 14, 27, 51], weight-space ensembling [48, 22], or ensemble distillation [19, 29]. For the adaptation of foundation models, several work propose to ensemble the fine-tuned and zero-shot models for better performance: Wortsman et al. [48] ensembles them in weight space; ProGrad and ProReg [55, 56] propose to fuse them via knowledge distillation. Our GLA is orthogonal to these approaches, as it concentrates on mitigating the biases in foundation models that are detrimental to ensemble models.

**Logit adjustment.** Logit adjustment [35, 45, 24, 49, 20] is a post-hoc technique to adjust the biased output of classification networks. Kang *et al* [24] proposes an element-wise scaling adjustment for the classifier weight. Tang *et al* [45] removes the projection of features on a global biased direction. Menon [35] derives the theoretically optimal adjustment from the training distribution. Unlike the those approaches which rely on a transparent training data or class distribution, our GLA can eliminate the class bias without accessing to the pre-training statistics.

## 3 Methods

### 3.1 Setup

**Task.** Consider a classification problem with instances $\mathbf{x} \in \mathcal{X}$ and labels $y \in \mathcal{Y} = [K] = \{1, ..., K\}$. We have a zero-shot model $f_{\mathsf{zs}}$ (given below), a downstream dataset $\mathcal{D}_s = \{\mathbf{x}_i, y_i\}_{i=1}^{N_s}$ drawn from source distribution $P_s$ and a fine-tuned model $f_{\mathsf{ft}}$ (given below) trained on the dataset $\mathcal{D}_s$. Give a model $f : \mathcal{X} \to \mathbb{R}^K$ that outputs prediction score, we define the risk of $f$ on the target distribution $P_t$ as the mis-classification rate: $\mathcal{R}_t(f) = \mathbb{E}_{\mathbf{x}, y \sim P_t}[y \neq \arg\max_i f(\mathbf{x})_i]$. Our goal is to learn a model $f_{\mathsf{gla}}$ that best leverages $f_{\mathsf{zs}}$ and $f_{\mathsf{ft}}$ that minimizes the risk $\mathcal{R}_t$.

**Zero-shot models.** We primarily explore CLIP [40] for zero-shot models. CLIP consists of a visual encoder $\Phi_{\mathsf{v}}(\mathbf{x})$ and a text encoder $\Phi_{\mathsf{t}}(\mathbf{t})$, producing $l_2$-normalized features from an image $\mathbf{x}$ and a text $\mathbf{t}$ respectively. Zero-shot model $f_{\mathsf{zs}}$ for $K$ classes is enabled by matching image features $\mathbf{v} = \Phi_{\mathsf{v}}(\mathbf{x})$ with classification weights $\mathbf{w}_k = \Phi_{\mathsf{t}}(\mathbf{t}_k)$, where $\mathbf{t}_k$ is obtained by extending the class name $\{c_k\}$ to a pre-defined prompt, *e.g.*, "a photo of a $\{c_k\}$.". Additional details are provided in Appendix B.2. The probability of $\mathbf{x}$ being classified as $y$ is defined as:

$$P(y|\mathbf{x}) = \text{softmax}(f_{\mathsf{zs}}(\mathbf{x}))_y = \frac{\exp(\mathbf{v}^T \mathbf{w}_y)}{\sum_{k=1}^{K} \exp(\mathbf{v}^T \mathbf{w}_k)}. \tag{1}$$

**Fine-tuned models.** Standard fine-tuning initializes the model $f_{\mathsf{ft}}$ with the pre-trained parameters and then solve $f_{\mathsf{ft}} = \arg\min_f \mathcal{R}_s(f)$ to minimize the risk on downstream dataset. We consider three common variants of fine-tuning: (1) end-to-end, where all parameters of $\Phi_{\mathsf{v}}$ and $\mathbf{w}_k$ are updated; (2) linear probing, where only $\mathbf{w}_k$ is modified while $\Phi_{\mathsf{v}}$ is fixed; (3) prompt tuning, where the text input $\mathbf{t}_k$ is learned, while keeping $\Phi_{\mathsf{v}}$ and $\Phi_{\mathsf{t}}$ freezed. See Appendix B.3 for details on fine-tuning methods.

**Notation.** Let $P_p(y)$, $P_s(y)$ and $P_t(y)$ be the marginal probability of class $y \in [K]$ for pre-training, source (training) and target (test) distribution, respectively. Let $\pi_p$ and $\pi_s$ denote the log probabilities of class for pre-training and training distribution, *i.e.*, $\pi_p(y) = \log P_p(y)$ and $\pi_s(y) = \log P_s(y)$.

### 3.2 Generalized Logit Adjustment Framework

Fine-tuned models often yield significant gains compared to zero-shot models, and ensembling them can further improve performance. This leads to a natural question: How should we best leverage

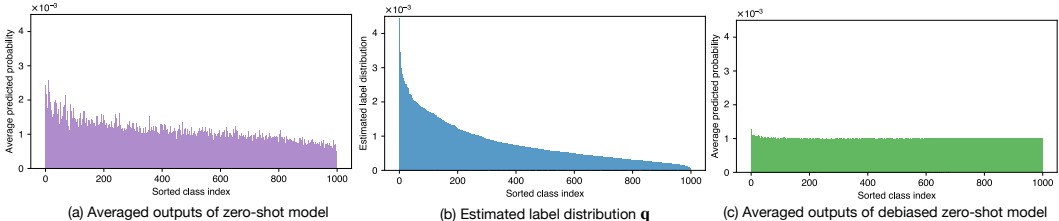

Figure 2: Illustration of debiasing process on ImageNet validation set. (a) The original distribution of zero-shot outputs; (b) the estimated pre-train distribution $\mathbf{q}$ based on our algorithm; (c) the distribution of debiased zero-shot outputs using estimated $\mathbf{q}$.

the zero-shot and fine-tuned models for the prediction tasks? We attempt to answer by proposing generalized logit adjustment in Definition 1.

**Definition 1.** *(GLA) The Generalized Logit Adjustment (GLA) model* $f_{\mathsf{gla}}$ *is defined as follows:*

$$f_{\mathsf{gla}}(\mathbf{x}) = f_{\mathsf{ft}}(\mathbf{x}) + f_{\mathsf{zs}}(\mathbf{x}) - \pi_s - \pi_p. \tag{2}$$

In Section 4.2, we prove that given the zero-shot and fine-tuned models, our GLA model is the *Bayes* optimal classifier and no other combination of the two models can outperform it. Here remains one important question: how could we obtain $\pi_p$ as we have no access to the pre-training statistics? We provide the estimation process of $\pi_p$ in Eq (4) and postpone the justification in Section 4.3. The entire GLA algorithm consists of two steps which is given as follows:

**Step 1: Estimation of** $\pi_p$. Let $\mathbf{q}$ be an arbitrary *probability simplex* over $K$ classes. Given the validation data from $P_t$ or the balanced training data, we can estimate the $\hat{\pi}_p = \log \mathbf{q}^*$ as the constrained optimization problem (proof in Section 4.3):

$$\mathbf{q}^* = \underset{\mathbf{q}}{\operatorname{argmin}} \, \mathcal{R}_t(f_{\mathsf{zs}} - \log \mathbf{q})$$
$$\text{s.t. } \mathbf{q}_i \geq 0, \text{for } i \in [K],$$
$$\sum_{i \in [K]} \mathbf{q}_i = 1. \tag{3}$$

We constrain the sum of $\mathbf{q}$ to be 1 and ensure that each element is non-negative, guaranteeing that it forms a valid probability distribution. We solve the following Lagrangian problem to find optimal $\mathbf{q}^*$:

$$\min_{\mathbf{q}} \max_{\lambda_i \geq 0, \upsilon} \mathcal{R}_t(f_{\mathsf{zs}} - \log \mathbf{q}) - \sum_i \lambda_i \mathbf{q}_i + \upsilon(1 - \sum_{i \in [K]} \mathbf{q}_i) \tag{4}$$

**Step 2: GLA ensembling.** Given the estimated $\hat{\pi}_p$ and the known downstream training $\pi_s$, we ensemble the zero-shot model $f_{\mathsf{zs}}$ and the fine-tuned model $f_{\mathsf{ft}}$ to get our GLA model $f_{\mathsf{gla}}$ via Eq. (2). We can regard $f_{\mathsf{zs}} - \hat{\pi}_p$ and $f_{\mathsf{ft}} - \pi_s$ as the debiased zero-shot model (Figure 2(c)) and debiased fine-tuned models, respectively. Our GLA is actually ensembling two debiased models. Note that, different from [55, 48], we *do not* require a hyper-parameter to adjust the contribution of the two models, the optimal solution is to combine them equally (see Section 4.2 for justification).

## 4 Theoretical Analysis

In this section, we explain why our GLA model best ensembles the zero-shot and fine-tuned models (Section 4.2) and justify the estimation process of the pre-training label distribution $\pi_p$ (Section 4.3). We start with some preliminaries on *Bayes* optimal classifier.

### 4.1 Preliminaries

Suppose we have a pair $(X, Y) \sim P$ takes values in $\mathcal{X} \times \mathcal{Y}$, where $Y$ is the class label of input $X$.

**Definition 2.** *The 0-1 error (risk) of a classifier* $\hat{y} : \mathcal{X} \to \mathcal{Y}$ *on distribution $P$ is given by:*

$$\mathcal{R}(\hat{y}) = P(Y \neq \hat{y}(X)) \tag{5}$$

However, the 0-1 error is non-smooth, one typically minimizes a surrogate loss $\ell$, *e.g.*, cross-entropy: $\ell(f(\mathbf{x}), y) = \log[\sum_{i \in [K]} \exp(f(\mathbf{x})_i - f(\mathbf{x})_y)]$, where $\hat{y}(\mathbf{x}) = \arg\max_i f(\mathbf{x})_i$. It is known that the cross-entropy loss is *Bayes consistent* [53], *i.e.*, a nearly optimal minimizer of the cross-entropy loss $(\mathbb{E}_{\mathbf{x},y \sim P}[\ell(f(\mathbf{x}), y)]$ is also a nearly optimal optimizer of the mis-classification error $(\mathbb{E}_{\mathbf{x},y \sim P}[y \neq \hat{y}(\mathbf{x})])$.

**Definition 3.** *The Bayes optimal classifier $y^*$ for $P$ given input $\mathbf{x}$ is defined as:*

$$y^*(\mathbf{x}) = \underset{y \in \mathcal{Y}}{\arg\max}\, P(y|\mathbf{x}) \tag{6}$$

It is called *Bayes* optimal classifier because on the average *no* other classifier using the same hypothesis and prior knowledge can outperform it.

**Lemma 1.** *The Bayes optimal classifier $y^*$ for $P$ has lower risk than all classifiers $\hat{y} : \mathcal{X} \to \mathcal{Y}$.*

$$\mathcal{R}(y^*) \leq \mathcal{R}(\hat{y}) \tag{7}$$

## 4.2 Generalized Logit Adjustment Leads to Better Ensembling

**Zero-shot and fine-tuned models are complementary.** We revisit an empirical phenomena observed in [48] Section 5.1: After exploring a series of measures of diversity, covering predictions and features, they find that zero-shot and fine-tuned models have diverse predictions, despite sharing the same backbone. As the two data distribution $P_s$ and $P_p$ is known to be different, the resulting models leverage different cues to predict: fine-tuned models risk exploiting *spurious correlations and in-domain patterns* which only hold for downstream dataset [46, 3]; On the other hand, zero-shot CLIP models capture *stable correlations* across diverse domains and exhibit much higher robustness [40]. For instance, zero-shot models rely on robust features for decisions that can achieve high performance on sketch and adversarial samples, while the fine-tuned models that trained on real images typically fail on these samples, as they rely on spurious correlations that only hold on real images. We formulate the phenomena in the following assumption.

**Assumption 1.** *Zero-shot and fine-tuned models have diverse predictions:*

$$(f_{\mathsf{ft}}(\mathbf{x}) \perp f_{\mathsf{zs}}(\mathbf{x}))|y. \tag{8}$$

We derive the conditional probability $P_t(y|f_{\mathsf{ft}}(\mathbf{x}), f_{\mathsf{zs}}(\mathbf{x}))$ w.r.t. the outputs of $f_{\mathsf{zs}}(\mathbf{x})$ and $f_{\mathsf{ft}}(\mathbf{x})$:

**Lemma 2.** *For a balanced target distribution[1], where $P_t(y) = 1/K$ for all $y \in [K]$, we have:*

$$P_t(y|f_{\mathsf{ft}}(\mathbf{x}), f_{\mathsf{zs}}(\mathbf{x})) = \mathrm{softmax}(\underbrace{f_{\mathsf{ft}}(\mathbf{x}) + f_{\mathsf{zs}}(\mathbf{x}) - \pi_s - \pi_p}_{f_{\mathsf{gla}}(\mathbf{x})})(y) \tag{9}$$

Intuitively, since the zero-shot and fine-tuned models provide diverse predictions, conditioned on the two predictions is equivalent to adding the logits in log space. Additionally, as the target distribution is class-balanced, we need to remove the class bias of two models by subtracting $\pi_s$ and $\pi_p$. The formal proof is given in Appendix A.1. Note that the RHS of Eq. (9) is exactly the softmax output of our GLA model by Definition 1, which exhibits the following property:

**Proposition 1.** *Let $g : \mathbb{R}^K \times \mathbb{R}^K \to \mathbb{R}^K$ be an arbitrary function that ensemble the outputs of $f_{\mathsf{zs}}$ and $f_{\mathsf{ft}}$. Our GLA classifier $f_{\mathsf{gla}}$ has lower risk than any function $f_g(\mathbf{x}) = g(f_{\mathsf{zs}}(\mathbf{x}), f_{\mathsf{ft}}(\mathbf{x}))$, i.e.*

$$\mathcal{R}_t(f_{\mathsf{gla}}) \leq \mathcal{R}_t(f_g). \tag{10}$$

*Proof.* From Lemma 2 and Definition 1, we have:

$$\underset{y \in \mathcal{Y}}{\arg\max}\, f_{\mathsf{gla}}(\mathbf{x})_y = \underset{y \in \mathcal{Y}}{\arg\max}\, \mathrm{softmax}(f_{\mathsf{ft}}(\mathbf{x}) + f_{\mathsf{zs}}(\mathbf{x}) - \pi_s - \pi_p)_y = \underset{y \in \mathcal{Y}}{\arg\max}\, P_t(y|f_{\mathsf{ft}}(\mathbf{x}), f_{\mathsf{zs}}(\mathbf{x})),$$
$$\tag{11}$$

which means $f_{\mathsf{gla}}$ is the Bayes optimal classifier (see Definition 3) given $f_{\mathsf{ft}}(\mathbf{x})$ and $f_{\mathsf{zs}}(\mathbf{x})$. According to Lemma 1, any other classifier $g(f_{\mathsf{ft}}(\mathbf{x}), f_{\mathsf{zs}}(\mathbf{x}))$ must have higher risk, *i.e.*, $\mathcal{R}_t(f_{\mathsf{gla}}) \leq \mathcal{R}_t(f_g)$. $\quad\square$

---

[1]This lemma can be easily extended to imbalanced target distributions (proof in Appendix A). Yet, as most test sets are class-balanced, we focus on the balanced case for brevity.

Proposition 1 demonstrates that our $f_{\text{gla}}$ model is the *best* model, as it has the lowest risk on target distribution. Proposition 1 further explains the superiority of $f_{\text{gla}}$ over the fine-tuned model $f_{\text{ft}}$ and the naive ensemble $f_{\text{ens}}(\mathbf{x}) = f_{\text{ft}}(\mathbf{x}) + f_{\text{zs}}(\mathbf{x})$:

**Corollary 1.** $f_{\text{gla}}$ *performs better than fine-tuned model $f_{\text{ft}}$ and naive emsembling $f_{\text{ens}}$:*

$$\mathcal{R}_t(f_{\text{gla}}) \leq \mathcal{R}_t(f_{\text{ft}}), \ \mathcal{R}_t(f_{\text{gla}}) \leq \mathcal{R}_t(f_{\text{ens}}) \tag{12}$$

**Discussion: when do the GLA models degenerate?** Note that there are two equality signs in Eq. (12), indicating that the performance of the GLA model can degenerate to be equivalent to that of the fine-tuned model and naive ensembling in the following two cases.

**Case 1**: For the first equality, if zero-shot model $f_{\text{zs}}(\mathbf{x})$ provides no further information about $y$ given $f_{\text{ft}}(\mathbf{x})$, *i.e.*, $(y \perp f_{\text{zs}}(\mathbf{x}))|f_{\text{ft}}(\mathbf{x})$, then $P_t(y|f_{\text{ft}}(\mathbf{x}), f_{\text{zs}}(\mathbf{x}))$ degenerates to $P_t(y|f_{\text{ft}}(\mathbf{x}))$ and the first equality applies. However, in practice, as downstream model and zero-shot model provides diverse predictions, we usually encounter strict inequality, *i.e.*, $\mathcal{R}_t(f_{\text{gla}}) < \mathcal{R}(f_{\text{ft}})$.

**Case 2**: The second equality applies when pre-training and downstream training distribution are both class-balanced. In fact, the pre-training dataset for foundation models are known to be highly skewed. Therefore, in most cases, we have $\mathcal{R}_t(f_{\text{gla}}) < \mathcal{R}_t(f_{\text{ens}})$.

In summary, the above two equalities are usually unattainable, which means that theoretically, our GLA model performs better than both the fine-tuned and the naive ensemble models.

### 4.3   Estimate the label bias of the pre-training dataset

However, $\pi_p$ is usually unknown as we have no access to pre-training dataset. In this work, we seek to estimate $\pi_p$ using the zero-shot models and the downstream data. Similar to Proposition 1, we have the following proposition says that $f_{\text{zs}} - \pi_p$ has lower error on target distribution than any other classifiers that use $f_{\text{zs}}$, see Appendix A.2 for the full proof.

**Proposition 2.** *Let $h : \mathbb{R}^K \to \mathbb{R}^K$ be an arbitrary function that predicts labels using the outputs of the zero-shot model $f_{\text{zs}}(\mathbf{x})$. Let the derived classifier be denoted as $f_h(\mathbf{x}) = h(f_{\text{zs}}(\mathbf{x}))$. The classifier $f_{\text{zs}} - \pi_p$ is better than any $f_h(\mathbf{x})$: $\mathcal{R}_t(f_{\text{zs}} - \pi_p) \leq \mathcal{R}_t(f_h(\mathbf{x}))$.*

Let $\mathbf{q}$ be an arbitrary probability simplex over $K$ classes, then we have $\mathcal{R}_t(f_{\text{zs}}(\mathbf{x}) - \pi_p) \leq \mathcal{R}_t(f_{\text{zs}}(x) - \log \mathbf{q})$. Therefore, we choose to optimize a *probability simplex* $\mathbf{q}$ over $K$ classes such that the model $f_{\text{zs}} - \log \mathbf{q}$ achieves the minimal empirical risk, as formulated in Eq. (3) (the Step 1 of GLA algorithm). Once we obtain the estimated class prior $\hat{\pi}_p = \log \mathbf{q}$, we can easily implement the GLA model by ensembling $f_{\text{gla}}(\mathbf{x}) = f_{\text{ft}}(\mathbf{x}) + f_{\text{zs}}(\mathbf{x}) - \pi_s - \hat{\pi}_p$ (the Step 2 of GLA algorithm).

**Toy experiment.** We conducted an experiment to show that the estimated label distribution closely approximates the true one. Specifically, we trained a model with a ResNet32 backbone on the imbalanced CIFAR-10-LT [10] dataset with an imbalanced ratio of 10. Subsequently, we used only the test set combined with our proposed method to estimate the label distribution. This procedure simulates scenarios where only downstream data is available and the pre-training data is inaccessible.

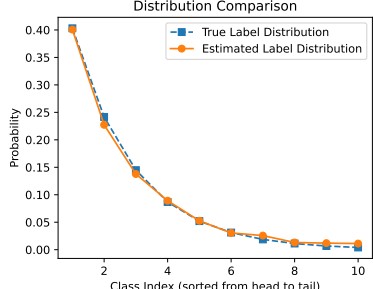

Figure 3 reveals a strong alignment between the estimated (orange line) and the actual distributions (blue line), which is further emphasized by a small KL-divergence value of 0.00062. The toy experiment validates the effectiveness of our debiasing method.

Figure 3: Estimating label bias of CIFAR-10-LT-IB-10.

**Discussion.** The $\log \mathbf{q}$ we estimated is not the marginal log-probability of the entire pre-training distribution but the label bias matches the downstream distribution. In the above toy experiment, although training and test sets show different label distributions, their conditional distribution $P(\mathbf{x}|y)$ remains invariant. In this case, our estimate will converge to the actual training label bias. For CLIP models, with diverse pre-training data, some might not align with the downstream domain, potentially compromising the accuracy of the estimation of the entire pre-training distribution.

However, we'd like to point out that removing the label bias of entire pre-training distribution may not optimal for downstream tasks. As a thought experiment, consider a pre-training dataset "sketch"

| Method | B/32 | B/16 |
|--------|------|------|
| Zero-shot | 63.2 | 68.3 |
| LP | 75.8 | 79.9 |
| E2E | 76.3 | 81.3 |
| WiSE-FT | 76.9 | 81.7 |
| **GLA** | **77.9** | **82.8** |

(a) ImageNet

| Method | B/32 | B/16 |
|--------|------|------|
| Zero-shot | 64.2 | 67.2 |
| LP | 80.5 | 83.1 |
| E2E | 89.4 | 91.0 |
| WiSE-FT | 90.0 | 91.3 |
| **GLA** | **90.7** | **91.9** |

(b) CIFAR100

| Method | B/32 | B/16 |
|--------|------|------|
| Zero-shot | 59.7 | 64.4 |
| LP | 77.1 | 83.1 |
| E2E | 83.7 | 90.2 |
| WiSE-FT | 84.5 | 90.7 |
| **GLA** | **85.0** | **91.1** |

(c) Stanford Cars

| Method | B/32 | B/16 |
|--------|------|------|
| Zero-shot | 62.2 | 64.8 |
| LP | 75.6 | 78.0 |
| E2E | 80.0 | 82.4 |
| WiSE-FT | 80.8 | 82.9 |
| **GLA** | **81.2** | **83.4** |

(d) SUN397

Table 1: Accuracy of various methods using CLIP ViT-B/32 and ViT-B/16. LP: linear probe; E2E: end-to-end fine-tuning. Results were obtained using the official implementation from WiSE-FT [48].

and "photo" styles for "dog" and "cat" samples. Suppose the sample size of "dog" and "cat" is equal but there are more "sketch dogs" than "sketch cats". This means that even if the overall distribution is balanced, each style isn't, resulting in biased zero-shot predictions. if we aim to deploy models for the "sketch dogs and cats" domain, adjusting the overall label bias is insufficient. Instead, the optimal label bias should be estimated on the "sketch" distribution. We also provide experiments using LAION-400M dataset in Appendix C.2, illustrating the situation when the downstream data diverges from the pre-training set.

## 5 Experiments

We evaluate our GLA on three real-world scenarios: many-shot (Section 5.1), few-shot (Section 5.2) and long-tail learning (Section 5.3). We show that our GLA boosts performance on all three settings.

### 5.1 Many-shot learning

**Datasets.** We use ImageNet [11] and CIFAR100 [26] for generic object classification, Stanford-Cars [25] for fine-grained classification, and SUN397 [50] for scene recognition. See Appendix B.1 for details.

**Baselines.** We compare GLA against four methods: (1) Zero-shot model, (2) Linear Probing (LP), (3) End-to-End fine-tuning (E2E), and (4) weight ensembling method WiSE-FT[48].

**Implementation details.** We consider two models: CLIP ViT-B/32 and ViT-B/16. For learning-based models, we fine-tune with AdamW using a cosine annealing learning rate scheduler. We fine-tune for 10 epochs on ImageNet and 20 epochs on other datasets. See Appendix B.3 for further details.

**Main results.** Table 1 compares our GLA with various baselines. We observe that our GLA can increase the performance of end-to-end fine-tuned models: it achieves $1.5\%$ gains on ImageNet. Compared to WiSE-FT, GLA gains $1.1\%$ top-1 accuracy boost on ImageNet dataset,. Beyond generic object recognition, our method also improves accuracy on the fine-grained dataset (Stanford Cars) and the scene recognition dataset (SUN397), by $0.4\%$ and $0.5\%$, respectively.

**Breakdown performance analysis.** To analyze the impact of pre-training label bias on fine-tuned and ensemble models, we present the breakdown results on ImageNet using CLIP ViT-B/16, as shown in Table 2. Specifically, we sort the class index using the estimated $\pi_p$, and assign the top third of the classes as the head classes, the last third as the tail classes, and the remaining classes as the medium classes. Due to the label bias, the zero-shot tail performance is significantly lower than the head one ($57.2\%$ *vs.* $78.0\%$). The resulting E2E models are also affected by the bias, with the tail performance being $6.3\%$ lower than the head. Existing ensemble WiSE-FT overlooks the bias, exhibiting noticeable degradation on the tail performances ($-0.9\%$) compared to E2E model, while our GLA stands out for all three subgroups.

**Estimated $\pi_p$ is transferable across different zero-shot models.** The estimated $\pi_p$ should be transferable across different zero-shot models if they are trained on the same pre-training dataset. To verify this, we employed a CLIP ViT-B/32 based zero-shot model to estimate $\pi_p$, which is subsequently used to debias zero-shot models based on CLIP ViT-B/16 and ViT-L/14. As shown in Table 3, our debiased models outperform the original zero-shot versions by a clear margin.

| Method | Head | Med. | Tail | All |
|---|---|---|---|---|
| Zero-shot | 78.0 | 69.8 | 57.2 | 68.3 |
| E2E | 83.6 | 83.0 | 77.3 | 81.3 |
| WiSE-FT | **85.3** | 83.7 | 76.4 | 81.7 |
| GLA | 85.2 | **84.3** | **78.8** | **82.8** |

Table 2: Breakdown results on ImageNet.

| Model | Source | Target | |
|---|---|---|---|
| | ViT-B/32 | ViT-B/16 | ViT-L/14 |
| $f_{zs}(\mathbf{x})$ | 63.4 | 68.8 | 75.6 |
| $f_{zs}(\mathbf{x}) - \hat{\pi}_p$ | **65.4** | **69.3** | **76.3** |

Table 3: Estimated $\hat{\pi}_p$ is transferable across different backbones. $\hat{\pi}_p$ is estimated using CLIP ViT-B/32.

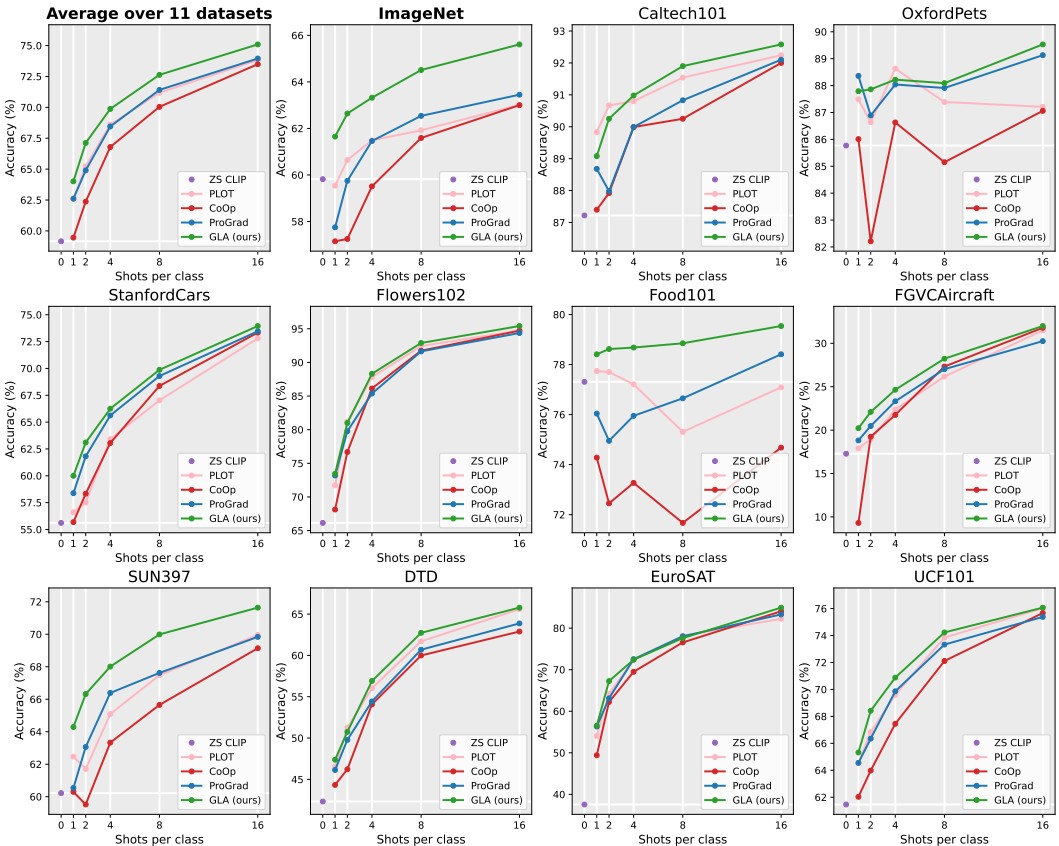

Figure 5: Accuracy (%) of few-shot learning on 11 datasets.

**Ensembling with mixing coefficient.** In Section 5.1, we prove the optimal solution is to combine the debiased zero-shot and fine-tuned models equally. We now examine the claim by introducing a mixture coefficient $\alpha \in [0, 1]$. The ensemble predictions are given by: $f_{gla}(\mathbf{x}, \alpha) = (1 - \alpha) \cdot (f_{zs}(\mathbf{x}) - \pi_p) + \alpha \cdot (f_{ft}(\mathbf{x}) - \pi_s)$. We compare the GLA and the naive ensembling with mixture $\alpha$ in Figure 4, where GLA meets its optimal performance at $\alpha = 0.5$, which is in line with our theoretical analysis. We also observe that the debiased zero-shot model increases accuracy by $2.3\%$ and our GLA consistently outperforms naive ensembling with various $\alpha$.

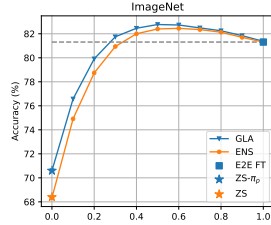

Figure 4: Accuracy with mixing coefficient $\alpha$.

## 5.2 Few-shot learning

For few-shot scenarios, we primarily choose prompt tuning for fine-tuning, since it is empirically more effective than end-to-end fine-tuning and linear probing [54, 55, 8].

**Datasets.** We follow CoOp[54] to use 15 datasets: ImageNet [11], Caltech101 [13], OxfordPets [37], StanfordCars [25], Flowers102 [36], Food101 [6], FGVCAircraft [34], EuroSAT [16], UCF101 [44],

| Method | Source | Target | | | |
|--------|--------|--------|--------|--------|--------|
| | IN | V2 | S | A | R |
| CLIP | 59.8 | 52.8 | 35.5 | 22.8 | 60.6 |
| CoOp | 61.9 | 54.3 | 32.5 | 21.8 | 54.2 |
| PLOT | 63.0 | 55.1 | 33.0 | 21.9 | 55.6 |
| ProGrad | 63.5 | 55.4 | 33.1 | 21.3 | 55.2 |
| GLA | **65.6** | **57.1** | **36.4** | **23.0** | **62.1** |

Table 4: Evaluation on robustness to distribution shift at 16 training shots.

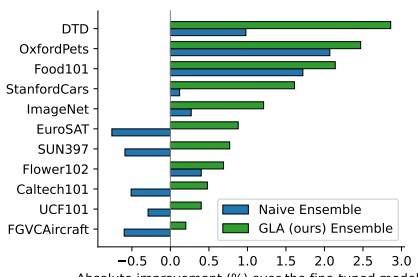

Figure 6: Comparison with naive ensembling.

| Model | IN. | Caltech. | Pets. | Cars. | Flowers. | Food. | Aircraft. | SUN. | DTD | EuroSAT | UCF. | Avg. |
|-------|-----|----------|-------|-------|----------|-------|-----------|------|-----|---------|------|------|
| $f_{\mathsf{zs}}$ | 59.8 | 87.1 | 85.8 | 55.6 | 65.3 | 77.9 | 17.1 | 60.3 | 42.3 | 37.5 | 61.5 | 59.1 |
| $f_{\mathsf{zs}} - \hat{\pi}_p$ | 62.3 | 89.9 | 85.9 | 57.6 | 67.2 | 78.6 | 18.4 | 63.5 | 43.0 | 39.0 | 62.4 | 60.7 |
| $\Delta$ | +2.5 | +2.7 | +0.1 | +2.0 | +1.9 | +0.7 | +1.3 | +3.2 | +0.7 | +1.4 | +0.9 | +1.6 |

Table 5: Comparision between zero-shot models ($f_{\mathsf{zs}}$) and debiased zero-shot models ($f_{\mathsf{zs}} - \hat{\pi}_p$).

DTD [9], SUN397 [50], ImageNet-{V2 [41], Sketch [47], A [18], R [17]}. We randomly select {1, 2, 4, 8, 16} shots for training and use the original test set for evaluation. See Appendix B.1 for details.

**Baselines.** We compare with three prompt tuning methods: (1) CoOp [54] optimizes prompts via empirical risk minimization; (2) ProGrad [55] prevents forgetting the general knowledge using zero-shot predictions; (3) PLOT [8] applies optimal transport to match the vision and text modalities.

**Implementation details.** We implement our proposed GLA method using CLIP-ResNet-50 as the foundation model and adopt class-specific prompt tuning from CoOp. Results are averaged over three seeds, with training configurations aligned with CoOp. See Appendix B.4 for further information.

**Main results.** Figure 5 summarizes the few-shot results on 11 datasets. The detailed accuracy and standard deviation are in Appendix C.1. Overall, our GLA clearly outperforms the baselines on average performance by a large margin, *e.g.*, our GLA gains 4.6%, 4.8%, 3.1%, 2.6% and 1.6% performance boost over CoOp at $1, 2, 4, 8, 16$ shots. In particular, on ImageNet dataset, we observe a large improvement, *e.g.*, $3.9\%, 2.9\%, 1.9\%, 2.0\%$ and $2.2\%$ over ProGrad at $1, 2, 4, 8, 16$ shots. Furthermore, on OxfordPerts, Food101 and SUN397 datasets, our method's performance remains stable and consistently improves with increasing sample sizes, while the one of the baseline methods fluctuates significantly. In particular, on Food101, baseline models even underperform zero-shot models by a large margin, while our GLA shows clearly better performance than zero-shot models.

**Robustness to distribution shifts.** Following CoOp, we used the ImageNet at 16 training shots as the source domain and assess the robustness on ImageNet-{V2, Sketch, A, R} datasets. Table 4 summarizes the results, where the prompt tuning baselines perform worse on distribution shifted datasets compared to the zero-shot model, as fine-tuning on limited data misleads the model to learn in-distribution correlations. In comparison, our GLA approach makes the best of both fine-tuned and zero-shot models, thus consistently outperforms other methods on both source and target domains.

**GLA improves accuracy over naive ensemble.** Figure 6 compares the results between our GLA and naive ensembling. We rank the absolute improvements over fine-tuning baseline at 16 training shots. In summary, our GLA demonstrates superior accuracy gains. It is worth noting that the naive ensembling does not always lead to improvements, *e.g.*, on EuroSAT, SUN, Caltech, UCF and Aircraft, naive ensembling even underperforms fine-tuning baseline.

**Debiased zero-shot models perform better.** We estimate $\pi_p$ at 16 shots and compare the original zero-shot models with the debiased zero-shot models in Table 5. It is clear that the debiasing leads to improvement on all 11 datasets, showcasing an average accuracy gain of $1.6\%$.

### 5.3 Long-tail learning

**Datasets and metrics.** We evaluate our method on two standard benchmarks: Places365-LT and ImageNet-LT [31]. In addition to top-1 accuracy, we report the accuracy on three test subsets

|  | Method | Many | Med | Few | All |
|---|---|---|---|---|---|
| LP | ERM | 64.4 | 39.8 | 20.1 | 46.6 |
| LP | LA | 56.7 | 50.5 | 46.4 | 52.3 |
| LP | BS | 57.7 | 50.5 | 43.9 | 52.4 |
| LP | **GLA** | **65.5** | **60.8** | **57.7** | **62.2** |
| PT | ERM | 66.6 | 57.7 | 53.8 | 60.6 |
| PT | LA | 67.9 | 60.0 | 58.7 | 62.9 |
| PT | BS | 67.0 | 62.2 | 59.4 | 63.5 |
| PT | **GLA** | **70.7** | **66.5** | **64.3** | **67.8** |
| E2E | ERM | 75.5 | 56.0 | 36.2 | 60.8 |
| E2E | LWS [24] | 70.4 | 62.6 | 56.2 | 64.7 |
| E2E | LA [35] | 70.4 | 62.8 | 56.9 | 64.9 |
| E2E | BS [43] | 71.2 | 62.5 | 56.8 | 65.1 |
| E2E | WiSE-FT [48] | 70.7 | 65.0 | 61.1 | 66.7 |
| E2E | BALLAD [33] | 71.0 | 66.3 | 59.5 | 67.2 |
| E2E | **GLA** | **72.1** | **66.4** | **63.4** | **68.2** |

(a) ImageNet-LT

|  | Method | Many | Med | Few | All |
|---|---|---|---|---|---|
| LP | ERM | 43.9 | 21.1 | 9.0 | 26.9 |
| LP | LA | 39.2 | 33.7 | 29.5 | 34.9 |
| LP | BS | 39.6 | 34.0 | 28.3 | 34.9 |
| LP | **GLA** | **43.6** | **40.2** | **38.8** | **41.1** |
| PT | ERM | 47.7 | 32.6 | 24.9 | 36.5 |
| PT | LA | 43.9 | 40.6 | 39.4 | 41.5 |
| PT | BS | 43.4 | 42.5 | 42.3 | 42.8 |
| PT | **GLA** | **47.0** | **47.1** | **47.8** | **47.2** |
| E2E | ERM | 46.3 | 27.4 | 12.5 | 31.3 |
| E2E | LWS [24] | 41.2 | 39.0 | 32.8 | 38.6 |
| E2E | LA [35] | 42.2 | 39.3 | 34.0 | 39.3 |
| E2E | BS [43] | 46.7 | 44.4 | 39.5 | 44.3 |
| E2E | WiSE-FT [48] | 46.0 | 44.0 | 43.9 | 44.7 |
| E2E | BALLAD [33] | **48.7** | 44.7 | 42.2 | 45.7 |
| E2E | **GLA** | 47.3 | **45.4** | **45.3** | **46.1** |

(b) Places365-LT

Table 6: The performances on ImageNet-LT and Places365-LT.

according to the number of samples per class: many-shot ($> 100$ samples), medium-shot ($20 \sim 100$ samples), and few-shot ($< 20$ samples). Detailed information is provided in Appendix B.1.

**Fine-tuning and long-tail learning baselines.** We compare our GLA with the combinations of fine-tuning protocols and long-tailed recognition methods. We consider three fine-tuning protocols: 1) Linear Probe (LP) 2) End-to-End (E2E) fine-tuning; 3) Prompt Tuning (PT). The three fine-tuning paradigms are introduced in Section 3.1 with details in Appendix B.3. We compare with 5 long-tail learning methods: 1) standard Empirical Risk Minimization (ERM); 2) Learnable Weight Scaling (LWS) [24]; 3) Logit Adjustment (LA) [35]; 4) Balanced Softmax (BS) [43], and 5) BALLAD [33], which is designed for VLMs. See Appendix B.6 for more details on long-tail baselines.

**Implementation Details.** For all combinations of the fine-tuning and long-tail learning baselines, visual backbones are initialized from CLIP-ResNet-50 and classifiers are initialized via zero-shot prompting. We use SGD for 50 epochs with batch size of 512. See Appendix B.6 for further details.

**Results.** Table 6 shows that our GLA method consistently surpasses baselines across all long-tailed datasets. Our approach outperforms PT-based models by $3.5\%$ and $4.4\%$ on ImageNet-LT and Places365-LT, respectively. Against E2E approaches, GLA exceeds not just the WiSE-FT but also the current SOTA method BALLAD, by a large margin, *e.g.*, 1pp gains on ImageNet-LT.

## 6 Conclusion and Limitation

In this paper, we identify the label bias in foundation models and underscore its adverse effects on downstream task performance. We propose the Generalized Logit Adjustment (GLA) framework for fine-tuning foundation models, which boosts the performance by effectively eliminating label bias and combining diverse predictions from zero-shot and fine-tuned models. We prove that when presented with zero-shot and fine-tuned models, our GLA is the Bayes optimal classifier for downstream task. Extensive experiments across a diverse range of tasks and fine-tuning framework demonstrate the effectiveness of our approach. We believe that the proposed GLA may partially improve the fairness and credibility of foundation models.

The first limitation is that we only focus on the label bias while other forms of model biases, *e.g.*, representation bias [5], cannot be addressed by our algorithm yet. The second limitation is that we primarily focus on enhancing the fine-tuning performance for discriminative models. However, applying our GLA framework to generative models presents challenges. For instance, language generation operates as a Markov process, meaning each output depends on previous ones. This implies it's not straightforward to estimate the biasedness of a sequence with our GLA, as we only compute the bias in a pre-defined and independent label space.

## Acknowledgments

This research is supported by the National Research Foundation, Singapore under its AI Singapore Programme (AISG Award No: AISG2-PhD-2021-01-002 and AI Singapore AISG2-RP-2021-022).

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

# A   Proofs

## A.1   Proof of Lemma 2

**Restated Lemma  (Lemma 2).** *Let $\pi_t$ denote the log probability of the target label distribution* $\pi_t(y) = \log P_t(y)$, *we have:*

$$P_t(y|f_{\text{ft}}(\mathbf{x}), f_{\text{zs}}(\mathbf{x})) = \text{softmax}(f_{\text{ft}}(\mathbf{x}) + f_{\text{zs}}(\mathbf{x}) - \pi_s - \pi_p + \pi_t)(y). \tag{13}$$

*In class-balanced target distribution, Eq. (13) simplifies to:*

$$P_t(y|f_{\text{ft}}(\mathbf{x}), f_{\text{zs}}(\mathbf{x})) = \text{softmax}(f_{\text{ft}}(\mathbf{x}) + f_{\text{zs}}(\mathbf{x}) - \pi_s - \pi_p)(y). \tag{14}$$

*Proof.* Denote the output $\mathbf{e} = f_{\text{ft}}(\mathbf{x})$ and $\mathbf{z} = f_{\text{zs}}(\mathbf{x})$. We first use the Bayes Rule to decompose $P_t(y|\mathbf{e}, \mathbf{z})$ into $P_t(\mathbf{e}, \mathbf{z}|y)$, $P_t(y)$ and $P_t(\mathbf{e}, \mathbf{z})$ in Eq. (15), then rewrite $P_t(\mathbf{e}, \mathbf{z}|y)$ in Eq. (16) according to Assumption 1. Focusing on label shift problem [35, 20, 30] where $P(\mathbf{x}|y)$ does not change, we derive Eq. (17)

$$P_t(y|\mathbf{e}, \mathbf{z}) = \frac{P_t(\mathbf{e}, \mathbf{z}|y)P_t(y)}{P_t(\mathbf{e}, \mathbf{z})} \tag{15}$$

$$= P_t(\mathbf{e}|y)P_t(\mathbf{z}|y)\frac{P_t(y)}{P_t(\mathbf{e}, \mathbf{z})} \tag{16}$$

$$= P_s(\mathbf{e}|y)P_p(\mathbf{z}|y)\frac{P_t(y)}{P_t(\mathbf{e}, \mathbf{z})} \tag{17}$$

$$= \frac{P_s(y|\mathbf{e})P_s(\mathbf{e})}{P_s(y)}\frac{P_p(y|\mathbf{z})P_p(\mathbf{z})}{P_p(y)}\frac{P_t(y)}{P_t(\mathbf{e}, \mathbf{z})} \tag{18}$$

$$= \frac{P_s(y|\mathbf{e})}{P_s(y)}\frac{P_p(y|\mathbf{z})}{P_p(y)}\frac{P_s(\mathbf{e})P_p(\mathbf{z})P_t(y)}{P_t(\mathbf{e}, \mathbf{z})} \tag{19}$$

$$\tag{20}$$

Since $\mathbf{e}, \mathbf{z}$ are fixed, we can replace the terms that not rely on $y$ with a constant $C_1$ in Eq. (21). We replace $P_s(y) = \exp(\log P_s(y)) = \exp(\pi_s(y))$, $P_p(y) = \exp(\log P_p(y)) = \exp(\pi_p(y))$ and $P_t(y) = \exp(\log P_t(y)) = \exp(\pi_t(y))$. Suppose the underlying class-probabilities $P_s(y|\mathbf{e}) \propto \exp(\mathbf{e}_y)$ and $P_p(y|\mathbf{z}) \propto \exp(\mathbf{z}_y)$ for $y \in [K]$. Denote the constants $C_s$ and $C_p$ for normalizing $\exp(\mathbf{e}_y)$ and $\exp(\mathbf{z}_y)$ into probabilities, and merge all constants to $C = \frac{C_1}{C_s C_p}$, we get Eq. (23)

$$P_t(y|\mathbf{e}, \mathbf{z}) = \frac{P_s(y|\mathbf{e})}{P_s(y)}\frac{P_p(y|\mathbf{z})}{P_p(y)}P_t(y)C_1 \tag{21}$$

$$= \exp(\mathbf{e} + \mathbf{z} - \pi_s - \pi_p + \pi_t)(y)\frac{C_1}{C_s C_p} \tag{22}$$

$$= C \cdot \exp(\mathbf{e} + \mathbf{z} - \pi_s - \pi_p + \pi_t)(y) \tag{23}$$

Because the summation of $P_t(y|\mathbf{e}, \mathbf{z})$ is 1, $C = 1/\sum_{i \in [K]} \exp(\mathbf{e} + \mathbf{z} - \pi_s - \pi_p + \pi_t)(i)$. Therefore, we have:

$$P_t(y|f_{\text{ft}}(\mathbf{x}), f_{\text{zs}}(\mathbf{x})) = P_t(y|\mathbf{e}, \mathbf{z}) \tag{24}$$

$$= \frac{\exp(\mathbf{e} + \mathbf{z} - \pi_s - \pi_p)_y}{\sum_{i \in [K]} \exp(\mathbf{e} + \mathbf{z} - \pi_s - \pi_p + \pi_t)_i} \tag{25}$$

$$= \text{softmax}(f_{\text{ft}}(\mathbf{x}) + f_{\text{zs}}(\mathbf{x}) - \pi_s - \pi_p + \pi_t)_y \tag{26}$$

In class-balanced target distribution case, $\pi_t = \log \frac{1}{K}$ is constant. Since the softmax function is invariant to constant offsets, Eq. (26) simplifies to:

$$P_t(y|f_{\text{ft}}(\mathbf{x}), f_{\text{zs}}(\mathbf{x})) = \text{softmax}(f_{\text{ft}}(\mathbf{x}) + f_{\text{zs}}(\mathbf{x}) - \pi_s - \pi_p)_y \tag{27}$$

$$\square$$

| Dataset | Classes | Train size | Test size | Task |
|---|---|---|---|---|
| ImageNet | 1,000 | 1.28M | 50,000 | Object-level |
| CIFAR100 | 100 | 50,000 | 10,000 | Object-level |
| Caltech101 | 100 | 4,128 | 2,465 | Object-level |
| DTD | 47 | 2,820 | 1,692 | Textures |
| EuroSAT | 10 | 13,500 | 8,100 | Satellite images |
| FGVCAircraft | 100 | 3,334 | 3,333 | Fine-grained aircraft |
| Flowers102 | 102 | 4,093 | 2,463 | Fine-grained flowers |
| Food101 | 101 | 50,500 | 30,300 | Fine-grained food |
| OxfordPets | 37 | 2,944 | 3,669 | Fine-grained pets |
| StanfordCars | 196 | 6,509 | 8,041 | Fine-grained car |
| SUN397 | 397 | 15,880 | 19,850 | Scene-level |
| UCF101 | 101 | 7,639 | 3,783 | Action |
| ImageNetV2 | 1,000 | - | 10,000 | Robustness to collocation |
| ImageNet-Sketch | 1000 | - | 50,889 | Robustness to sketch domain |
| ImageNet-A | 200 | - | 7,500 | Robustness to adversarial attack |
| ImageNet-R | 200 | - | 30,000 | Robustness to multi-domains |

Table 7: The detailed statistics of datasets for many-shot and few-shot learning.

## A.2 Proof of Proposition 2

**Restated Proposition (Proposition 2).** *Suppose that the target distribution $P_p$ is class-balanced. Let $h : \mathbb{R}^K \to \mathbb{R}^K$ be an arbitrary function that predicts labels using the outputs of the zero-shot model $f_{\mathsf{zs}}(\mathbf{x})$. Let the derived classifier be denoted as $f_h(\mathbf{x}) = h(f_{\mathsf{zs}}(\mathbf{x}))$. The classifier $f_{\mathsf{zs}} - \pi_p$ is better than any $f_h(\mathbf{x})$: $\mathcal{R}_t(f_{\mathsf{zs}} - \pi_p) \leq \mathcal{R}_t(f_h(\mathbf{x}))$.*

*Proof.* Denote the output $\mathbf{z} = f_{\mathsf{zs}}(\mathbf{x})$. Similar to Eq. (15)-Eq. (26), we have

$$P_t(y|\mathbf{z}) = \frac{P_t(\mathbf{z}|y)P_t(y)}{P_t(\mathbf{z})} \tag{28}$$

$$= \frac{P_p(\mathbf{z}|y)P_t(y)}{P_t(\mathbf{z})} \tag{29}$$

$$= \frac{P_p(y|\mathbf{z})}{P_p(y)} \frac{P_t(y)}{P_t(\mathbf{z})} \tag{30}$$

$$= \exp(\mathbf{z} - \pi_p)(y) / \sum_{i \in [K]} \exp((\mathbf{z} - \pi_p)(i)) \tag{31}$$

$$= \mathrm{softmax}(\mathbf{z} - \pi_p) = \mathrm{softmax}(f_{\mathsf{zs}}(\mathbf{x}) - \pi_p) \tag{32}$$

Therefore, we have:

$$\underset{y \in \mathcal{Y}}{\mathrm{argmax}}(f_{\mathsf{zs}}(\mathbf{x}) - \pi_p)_y = \underset{y \in \mathcal{Y}}{\mathrm{argmax}}\,\mathrm{softmax}(f_{\mathsf{zs}}(\mathbf{x}) - \pi_p)_y = \underset{y \in \mathcal{Y}}{\mathrm{argmax}}\,P_t(y|f_{\mathsf{zs}}(\mathbf{x})) \tag{33}$$

Again, using Lemma 1, any other classifier $f_h(\mathbf{x})$ has higher risk than $f_{\mathsf{zs}}(\mathbf{x}) - \pi_p$, *i.e.*, $\mathcal{R}_t(f_{\mathsf{zs}} - \pi_p) \leq \mathcal{R}_t(f_h(\mathbf{x}))$. $\square$

# B Experimental Details

## B.1 Dataset details

**Many-shot and few-shot datasets.** For many-shot learning, we use ImageNet, CIFAR100, Stanford-Cars and SUN397 datasets. For few-shot learning, we evaluate models on 15 datasets. The details of each dataset are presented in Table 7.

**Long-tail datasets.** We use two standard long-tail benchmarks: Places365-LT and ImageNet-LT [31]. The skewness of a long-tailed training set is typically represented by imbalanced ratio, which is defined as $N_{\max}/N_{\min}$. $N_{\max}$ ($N_{\min}$) denotes the largest (smallest) number of instances per class. A

| Dataset | Size of all classes | Size of many classes | Size of medium classes | Size of few classes | Size of training samples | Imbalanced ratio |
|---|---|---|---|---|---|---|
| Places365-LT | 365 | 131 | 163 | 71 | 62.5K | 996 |
| ImageNet-LT | 1000 | 385 | 479 | 136 | 186K | 256 |

Table 8: Details of long-tailed datasets.

larger imbalanced ratio means a more imbalanced training set. The test sets are divided into three splits: many-shot subset contains classes with $> 100$ images, medium-shot subset includes classes with $\geq 20 \ \& \leq 100$ images, and few-shot subset covers classes with $< 20$ images. Details are listed in Table 8.

## B.2 CLIP zero-shot

We use prompt ensembling of 80 prompts provided by CLIP [48] for ImageNet, CIFAR100, and Caltech101 to improve performance, *i.e.*, averaging the text embedding of many captions, *e.g.*., "a photo of a $\{c_k\}$." and "an image of a $\{c_k\}$.". For OxfordPets, StanfordCars, Flowers102, Food101, FGV-CAircraft, EuroSAT, UCF101, DTD and SUN397, we use the pre-defined prompt from CoOp [54].

## B.3 Fine-tuned models

**End-to-end and linear probe fine-tuning.** We follow WiSE-FT [48] to implement fine-tuning. We initialize the classifier with the zero-shot classifier and the output of the image encoder $\Phi_v$ is normalized during fine-tuning. We fine-tune for a total of 10 epochs using AdamW [32] optimizer with default hyper-parameters $\beta_1 = 0.9, \beta_2 = 0.999, \epsilon = 10^{-8}$ and weight decay 0.1. We choose a batch size of 512. We use the same data augmentation and cosine-annealing learning rate schedule as [48].

## B.4 Prompt tuning.

Prompt tuning like CoOp [54] automates prompt engineering by learning the prompt given few samples from downstream tasks. CoOp provides two options of prompt design: unified prompt that is shared among all classes and class-specific prompt that is different for each class. In this paper, we adopt the class-specific prompt design as the fine-tuned model to implement GLA . In specific, given the word embedding $\mathbf{t}_k^0$ initialized by zero-shot prompts, we aim to learn a collection of class-specific word embedding $\mathbf{R} = \{\mathbf{r}_k\}_{k=1}^K$, such that the text input $\mathbf{t}_k = \mathbf{t}_k^0 + \mathbf{r}_k$ minimizes the empirical risk: $\mathbf{R}^* = \arg\min_{\mathbf{R}} \mathbb{E}_{\mathbf{x},y}[y \neq \arg\max_i f(x; \mathbf{R})_i]$.

We adhere CoOp to use CLIP ResNet-50 as image encoder for few-shot classification. The word embedding $\mathbf{R}$ is initialized from zeros. For the $m$ few-shot classification setting (where $m \in \{1, 2, 4, 8, 16\}$), we randomly sample $m$ training and $m$ validation points from the respective full datasets. For all few-shot datasets except ImageNet, the training epoch is set to 200 for 16/8 shots, 100 for 4/2 shots, and 50 for 1 shot. For ImageNet, the epoch is set to 50 for all shots. We fine-tune the prompt with SGD optimizer decayed by the cosine annrealing rule. The base initial learning rate and batch size are set to $10^{-4}$ and 32. When given an $m$-shot sample setting, we increase the learning rate and batch size by $m$ times simultaneously to accelerate the training speed.

## B.5 Estimation of the class prior

To estimate the log-probability of the pre-training distribution $\hat{\pi}_s = \log \mathbf{q}$, we utilize the optimization toolkit Cooper [15] from `https://github.com/cooper-org/cooper`. $\mathbf{q}$ is initialized as a uniformed distribution, $\mathbf{q}(y) = \frac{1}{K}$ for all $y \in [K]$. We use the standard SGD as the primal and dual optimizers for 2000 steps.

## B.6 Long-tail learning baselines and training details

We compared with 5 long-tailed classification methods:

1. Standard ERM: We learn the model by standard empirical risk minimization on the long-tailed data.

2. Learnable Weight Scaling (LWS) [24]: We first learn the model by standard ERM, then fix the model and learn to re-scale the magnitude of the classifier using class-balanced sampling.

3. Logit Adjustment (LA) [35]: We first learn the model by standard ERM, then compensates the long-tailed distribution by subtracting a class-dependent offset to the model outputs.

4. Balanced Softmax (BS) [43] modifies the Softmax cross-entropy loss which explicitly accommodate the label distribution shift during optimization.

5. BALLAD [33] first fine-tunes the vision-language models via contrastive loss on long-tailed data, then freezes the backbone and finally employs an adapter to enhance the representations of tail classes with re-sampling strategies.

For all combinations of the fine-tuning baselines and long-tailed learning methods, visual backbones are initialized from CLIP-ResNet-50 and all classifiers are initialized by feeding prompt with class names to the text encoder. We use SGD for all experiments with a momentum of 0.9 for 50 epochs with batch size of 512. The initial learning rate is set to $1.6 \times 10^{-3}$ which is decayed by the cosine annealing rule. To mitigate explosive gradients, we use the warmup learning rate equals to $10^{-5}$ during the first epoch. For the sake of fairness in comparison, all hyper-parameters of baselines are carefully searched using grid search on the validation set.

## C  Additional Experiments

### C.1  Few-shot learning accuracy

We provide mean and standard deviation in Table 9 in for {1, 2, 4, 8, 16} shots on all 11 few-shot learning datasets.

| Dataset | 1 shot | 2 shots | 4 shots | 8 shots | 16 shots |
|---|---|---|---|---|---|
| ImageNet | $61.65 \pm 0.15$ | $62.64 \pm 0.01$ | $63.32 \pm 0.07$ | $64.51 \pm 0.09$ | $65.61 \pm 0.03$ |
| Caltech101 | $89.08 \pm 0.09$ | $90.25 \pm 0.25$ | $90.98 \pm 0.43$ | $91.90 \pm 0.21$ | $92.58 \pm 0.42$ |
| OxfordPets | $87.79 \pm 0.15$ | $87.86 \pm 0.21$ | $88.22 \pm 0.21$ | $88.09 \pm 0.27$ | $89.53 \pm 0.16$ |
| StanfordCars | $60.00 \pm 0.14$ | $63.10 \pm 0.42$ | $66.25 \pm 0.19$ | $69.87 \pm 0.09$ | $73.95 \pm 0.11$ |
| Flowers102 | $73.45 \pm 0.60$ | $81.00 \pm 0.46$ | $88.31 \pm 0.65$ | $92.89 \pm 0.46$ | $95.41 \pm 0.32$ |
| Food101 | $78.41 \pm 0.07$ | $78.62 \pm 0.07$ | $78.68 \pm 0.06$ | $78.85 \pm 0.19$ | $79.54 \pm 0.47$ |
| FGVCAircraft | $20.22 \pm 0.59$ | $22.09 \pm 0.37$ | $24.65 \pm 0.85$ | $28.23 \pm 0.44$ | $31.99 \pm 0.50$ |
| SUN397 | $64.29 \pm 0.19$ | $66.32 \pm 0.16$ | $68.01 \pm 0.08$ | $69.99 \pm 0.18$ | $71.64 \pm 0.21$ |
| DTD | $47.38 \pm 1.23$ | $50.75 \pm 1.46$ | $56.90 \pm 0.20$ | $62.73 \pm 0.80$ | $65.78 \pm 0.49$ |
| EuroSAT | $56.50 \pm 1.34$ | $67.26 \pm 3.58$ | $72.40 \pm 2.43$ | $77.59 \pm 1.84$ | $84.93 \pm 1.89$ |
| UCF101 | $65.32 \pm 0.17$ | $68.42 \pm 0.81$ | $70.88 \pm 0.50$ | $74.23 \pm 0.24$ | $76.07 \pm 0.03$ |

Table 9: GLA Accuracy (%) with standard deviation of few-shot learning on 11 datasets.

### C.2  Experiments on LAION-400M

To support our thought experiment in the discussion of Section 4.3, we use the Open-CLIP ViT-B/16 [21], the first 20k image in LAION-400M datasets and the bias estimation method proposed by [2] to estimate the expected logits across 8 classes: "dog", "cat", "squirrel", "tiger", "elephant", "horse", "pig" and "bird". The bias estimation proposed by [2] provides a good estimation of $\log P(\mathbf{x})$ over the labels under pre-training distribution. Our GLA estimates the label bias matches the downstream domain, we consider two downstream domain styles, *i.e.*, "photo" and "sketch" from DomainNet [38] dataset. For each domain, we randomly sampled 50 images for each class.

| Method | dog | cat | squirrel | tiger | elephant | horse | pig | bird |
|---|---|---|---|---|---|---|---|---|
| expected logits by [2] | 0.059 | 0.039 | 0.043 | 0.053 | 0.043 | 0.062 | 0.061 | 0.028 |
| $\pi_p$ by "photo" | 0.059 | 0.041 | 0.047 | 0.055 | 0.048 | 0.062 | 0.060 | 0.033 |
| $\pi_p$ by "sketch" | 0.059 | 0.043 | 0.063 | 0.061 | 0.067 | 0.042 | 0.047 | 0.056 |

Table 10: Comparison among different bias estimation using Open-CLIP-ViT-B/16.

We present the expected logits estimated by [2], along with the one calculated from "photo" and "sketch" downstream domain data in Table 10. Since the softmax is invariant to constant offsets, *i.e.*,

| Method | Sketch | Real Photo |
|---|---|---|
| Zero-shot Open-CLIP | 92.25 | 97.00 |
| Debiased by [2] | 89.50 | 97.50 |
| Debiased by GLA | 93.00 | 97.75 |

Table 11: Performance on sketch and real photo domain.

$\text{softmax}(\mathbf{x} + c) = \text{softmax}(\mathbf{x})$, we align the three label bias to yield the same logits on "dog" class by subtracting constant values. We observe that the label bias estimated by the "photo" domain align closely with [2] due to its close resemblance to the pre-trained domain. Conversely, the "sketch" image style, which significantly differs from the pre-training domain, results in a more pronounced deviation in the pre-trained label bias.

Additionally, we apply the three estimated label biases to debias the zero-shot model and evaluate the classification performance. The results are shown in Table 11, where the superiority of our method becomes evident on "sketch" domain (93.00% vs 92.25%). Applying the label bias from [2] on the "sketch" domain degrades the model's performance (from 92.25% to 89.50%). This is attributed to the overall pre-training label bias does not adequately reflect the bias specific to the"sketch" domain.

