# OpenReview forum: "Generalized Logit Adjustment: Calibrating Fine-tuned Models by Removing Label Bias in Foundation Models"
_NeurIPS.cc/2023/Conference — NeurIPS 2023 poster_

### Official Review · Reviewer_D4ZP · 2023-06-10

**Soundness:** 4 excellent
**Presentation:** 4 excellent
**Contribution:** 3 good
**Rating:** 7
**Confidence:** 4

**Summary:**

This paper addresses biases in foundation models, such as CLIP, which due to the imbalanced training datasets, are skewed towards frequent semantics. The authors propose a Generalized Logit Adjustment (GLA) method, an optimization-based approach for debiasing these models. Despite inherent challenges, the GLA method demonstrates significant improvements across multiple tasks and datasets, achieving 1.5 pp accuracy gains on ImageNet and large average improvement (1.4-4.6 pp) on 11 few-shot datasets. Notably, the GLA method does not require access to the pre-training dataset, making it practical for fine-tuning scenarios. The paper offers both theoretical justification and extensive empirical evidence for the proposed method.


**Strengths:**

1. **Solid Theoretical Analysis**: The paper provides robust theoretical justification for the Generalized Logit Adjustment (GLA) method. It formalizes the estimation of label bias as a constrained optimization problem, proving that GLA is a Bayes optimal classifier.

2. **Practical Solution**: The proposed method doesn't require access to the pre-training dataset, which makes it practical for fine-tuning scenarios. This approach is useful given privacy and copyright concerns associated with accessing the label distribution of pre-training datasets.

3. **Comprehensive Evaluation**: The paper uses a comprehensive benchmark for evaluation, considering three real-world settings and three fine-tuning paradigms. This diverse approach to evaluation adds credibility to the findings.

4. **Significant Performance Improvement**: The paper reports notable improvements across various tasks, demonstrating the efficacy of the proposed GLA method. The substantial gains across multiple datasets reinforce the value of the method.


**Weaknesses:**

1. **Limited Validation**: The paper only presents results with a limited set of tasks and models. While they used the CLIP model for the zero-shot setup, the validity and effectiveness of the GLA framework with other models (like BERT, GPT, etc.) have not been validated.

2. **Estimating Pre-training Bias**: The estimation of pre-training label bias is done with downstream data. However, this could be problematic if the downstream data distribution significantly differs from the pre-training data distribution, leading to incorrect bias estimation and sub-optimal performance.


**Questions:**


The only two questions are listed in weakness.

**Limitations:**

While it's a promising beginning to uncover the biases in foundational models, it's crucial to recognize that the exploration should not be confined to computer vision alone. It is necessary to extend the studies to other fields as well.

---

> ### Author Rebuttal · Authors · 2023-08-09
>
> **Q1** The validity and effectiveness of the GLA framework with NLP models.
>
> **A1** We appreciate the reviewer for the comment. Applying this algorithm to other tasks and fields is also our future research direction.  For now, we primarily focus is on enhancing the fine-tuning performance for discriminative models. The NLP zero-shot models are dominated by generative models, applying our GLA framework for those models presents challenges. For example, the language generation is a markov process, which makes each output depend on outputs of previous steps. Namely, it is not trivial to estimate the biasedness of a sequence by our GLA where we only computes the bias on a pre-defined and independent label space. We will discuss it in the revision version and leave it as a future work.
>
> ---
>
> **Q2** Estimating pre-training bias when the downstream data distribution significantly differs from the one of pre-training data.
>
> **A2** While we acknowledge the concern raised, it is crucial to understand that our work is primarily focused on addressing the label shift problem~(also known as long-tail problem). Yes, you are right in pointing out that our GLA paradigm might not be optimally effective when the downstream data distribution diverges significantly from the pre-training data distribution, e.g. some downstream samples $\mathbf{x}_s$ do not have density in pre-training domain ($P_p(\mathbf{x}_s) \approx 0$) . We recognize this especially in the case of niche domains such as medical data, which are scarcely represented in the pre-training stage. We believe that incorporating such downstream data into the foundation model training might be the only way to resolve the extreme case, which is beyond our scope.
>
> Our methodology, nevertheless, has demonstrated substantial efficacy in the presence of label shift and can effectively mitigate the impact of pre-training label bias in a broad array of scenarios.

---

> > ### Comment · Reviewer_D4ZP · 2023-08-14
> >
> > Thank the authors for the explanation. I have read the comments and rebuttal, I decide to maintain the score as 7.

---

### Official Review · Reviewer_egNQ · 2023-06-27

**Soundness:** 3 good
**Presentation:** 3 good
**Contribution:** 3 good
**Rating:** 4
**Confidence:** 5

**Summary:**

This paper studies the label bias in Foundation Models, like CLIP. Specifically, authors model the estimation of the label bias as a constrained optimization problem, and propose a Generalized Logit Adjustment (GLA) method to debias foundation models. Experimental results demonstrate the effectiveness of proposed method.

**Strengths:**

-  Show that the skewness of pre-training distribution affects the performance of downstream tasks, i.e., label bias of foundation models.
-  Propose a GLA method to ensemble the debiased zero-shot and fine-tuned models.
-  Extensive experimental results demostrate the superiority of proposed method over conventional fine-tuning.

**Weaknesses:**

- Authors should discuss the difference with existing label bias estimation methods, e.g. [1,2].
-  [3] also studied the label bias in CLIP. What's the difference?
- There exist some strong assumption for proposed method.

[1] Lipton Z, Wang Y X, Smola A. Detecting and correcting for label shift with black box predictors. In ICML, 2018: 3122-3130.
[2] Garg S, Wu Y, Balakrishnan S, et al. A unified view of label shift estimation. In NeurIPS, 2020, 33: 3290-3300.
[3] Wang X, Wu Z, Lian L, et al. Debiased learning from naturally imbalanced pseudo-labels. In CVPR, 2022: 14647-14657.

**Questions:**

- The Proposition 1 shows that $f_{gla}$ is the best model, or nearly optimal Bayesian classifer. Thus what leads to the error between the realistic results and the ideal performance of optimal Bayesian classifer.
- What's the source of the validation data for a specific problem. In all experiments, the compared  fine-tuning methods whether or not to use the validation data.
- Please illustrate that ''Intuitively, since the zero-shot and fine-tuned models provide diverse predictions, conditioned on the two predictions is equivalent to adding the logits in log space".

**Limitations:**

The proposed method can effectively deal with label bias in foundation models. However, it depends on some strong assumptions. E.g., the estimation relies on the validation data. How to guarantee that a good estimation can be obtained with poor validation data.

---

> ### Author Rebuttal · Authors · 2023-08-09
>
> **Q1** Discussion with existing label estimation methods[1][2].
>
> **A1** [1-2] and our GLA work under different scenarios and require different forms of input.
> [1-2] face a problem where the test prior is unknown but training data are given. In the context of foundation model adaptation, [1-2] require both \textbf{labeled pre-training data} and **unlabeled downstream test data** to estimate label shift by matching the moment of confusion matrices. However, the pre-training data is often inaccessible, which makes [1-2] impractical for foundation models. In contrast, the proposed GLA targets on the setting where the pre-training label prior is unknown but the test distribution is class-balanced.
> Our method estimates the label bias of pre-training data by adjusting the margin to produce a Bayes optimal classifier.
> Technically, we only have to access **downstream source data** for debiasing. We will add the above discussion in the revised version.
>
> ---
> **Q2** What's the difference with [3]?
>
> **A2** Our GLA converges to the true class-probability of pre-training data while the one estimated by DebiasedPL[3] is asymptotically biased to the downstream distribution.
>  Recall that the DebiasedPL[3] uses the moving average of the outputs of zero-shot model $P_\text{zs}(y|\mathbf{x})$ (predicted probability of downstream data) as the class-specific margin for debiasing.
> The  probability for class $y$ estimated by DebiasedPL[3] can be viewed as
> $\hat{p}(y)=\frac{1}{N} \sum_{i=1}^N  P_{zs}(y|\mathbf{x}_i)$, where $\mathbf{x}_i$ is the sample from downstream dataset $\mathcal{D}_s \sim P_s^N$.
>
> As $N \rightarrow \infty$, $\hat{p}(y) \rightarrow E_{\mathbf{x}\sim P_s}[P_{zs}(y|\mathbf{x})]$.
>
> However, the true class-probability of pre-training distribution should be $p(y)=E_{\mathbf{x} \sim P_p}[P_\textbf{zs}(y|\mathbf{x})]$. As long as $P_s \neq P_p$, i.e., pre-training distribution differs from downstream distribution, the class-specific margin estimated by [3] is biased. In contrast, In Proposition 2, we prove that $f_\text{zs}(\mathbf{x})-\log \mathbf{q}$ is the Bayes optimal classifier given $f_\text{zs}$ that achieves the lowest error on the target distribution. Our estimation of $\hat{\mathbf{q}}$ converges to the true one.
>
> Note that we are not intended to undermine the value of [3], as they focus on semi-supervised learning where the labels of training data are not available.
> In contrast, our GLA is given labeled fine-tuning data and able to estimate a more accurate label shift. We will add the discussion in the revised version.
>
> ---
> **Q3** There exist some strong assumption for proposed method. E.g., the estimation relies on the validation data. How to guarantee that a good estimation can be obtained with poor validation data?
>
> **A3** We are sorry that we did not emphasize the implementation details, leading to your misunderstanding.
> In all conducted experiments, **no validation set** was used for $\pi_p$ estimation. Recall that in line 126-127, we mentioned that we have two choices: ``we can use validation data or the balanced training data'' to estimate $\pi_p$. As the training samples are more abundant than validation samples, we choose to use training set rather than validation set. For many-shot and few-shot learning, as the training set is balanced at hand, we directly use it for $\pi_p$ estimation. For long-tailed learning, we re-balanced the training set by up-sampling to estimate $\pi_p$. For the sake of fairness in comparison, the validation set is only used to search hyper-parameters for all baselines and our methods.
>
> ---
> **Q4** What leads to the error between the realistic results and the ideal performance of optimal Bayesian classifier?
>
> **A4** Our Proposition 1 analyses the asymptotic behavior of GLA. In practice, GLA's performance hinges on the estimation of pre-training $\pi_p$, a factor influenced by the pre-trained model's generalization gap. From structural risk minimization theory[1-2], the test error
> $\delta_j=E_{\mathbf{x} \sim \hat{P}_j}[\ell(f(\mathbf{x},j)]$
>
> $- E_{\mathbf{x} \sim P_j}[\ell(f(\mathbf{x},j)] $
>
> for class $j$ is bounded by $C/\sqrt{n_j}$, where $n_j$ is the class size for $j$ and $C$ is a constant that reflects model complexity. When the sample size of less frequent classes is small during pre-training, the large generalization gaps might lead to the estimation error of $\pi_p$.
>
>     [1] Kaidi Cao, Colin Wei, Adrien Gaidon, Nikos Arechiga, and Tengyu Ma. Learning imbalanced datasets with label-distribution-aware margin loss. NeurIPS, 2019.
>
>     [2] Shiori Sagawa, Pang Wei Koh, Tatsunori B. Hashimoto, and Percy Liang. Distributionally robust neural networks for group shifts: On the importance of regularization for worst-case generalization. ICLR, 2019.
> ---
>
> **Q5** Illustrate that ''since the zero-shot and fine-tuned models provide diverse predictions, conditioned on the two predictions is equivalent to adding the logits in log space".
>
> **A5** The detailed explanation is included in Appendix Eq.(20-21).  Denote the output $\mathbf{e}=f_{ft}(\mathbf{x})$ and $\mathbf{z}=f_{zs}(\mathbf{x})$, then
>
> $P_t(y|\mathbf{e},\mathbf{z})=\frac{P_s(y|\mathbf{e})}{P_s(y)}\frac{P_p(y|\mathbf{z})}{P_p(y)}C_1,$
>
> where $C_1$ is a constant that not rely on $y$ (see line 323-324 for further explanation on $C_1$). We can rewrite $P_s(y)=\exp(\log P_s(y))=\exp(\pi_s(y))$ and  $P_p(y)=\exp(\log P_p(y))=\exp(\pi_p(y))$. As the underlying class-probability $P_s(y|\mathbf{e}) \propto \exp(\mathbf{e}_y)$ and $P_p(y|\mathbf{z}) \propto \exp(\mathbf{z}_y)$. The above equation equals to
>
> $P_t(y|\mathbf{e},\mathbf{z})=\frac{\exp(\mathbf{e}_y)\exp(\mathbf{z}_y)}{\exp(\pi_s(y))\exp(\pi_p(y))} C_1/C_2=\exp(\mathbf{e}+\mathbf{z}-\pi_s-\pi_p)_yC_1/C_2,$
>
> As the logits have to apply exponential function to get the probability, multiplying two probability equals to adding logits. We refer to ``log space'' as both logits and $\pi=\log p$ represent the logarithm of probability.

---

> > ### Comment · Reviewer_egNQ · 2023-08-11
> > **The concerns are not perfectly clarified**
> >
> > Thanks for your detailed response, and most of them are not perfectly clarified. Firstly, the assumption in which test distribution is class-balanced in A1 seems to be impractical. The more reasonable setting is that the test prior is unknown. Secondly, DebiasedPL is general and works in more scene, e.g., semi-supervised learning. Though proposed method has theoretical Bayes optimal classifier, the strong assumption and  limited improvement weaken the contribution. Thirdly, though no validation set are used to estimation $\pi_p$, the used balanced training data can not ensure the solution is optimal. E.g., the noisy label or imbalanced test distribution may deteriorate the performance. I tend to decrease my score to 4.
> >
> > Reviewer egNQ

---

> > > ### Author Response · Authors · 2023-08-11
> > > **Some misunderstandings**
> > >
> > > Thanks for your reply. We found some of your points to be unreasonable.
> > >
> > > Firstly, [1-2] require the test data to estimate the test prior. If our algorithm is provided with test data, we can directly use Eq.(3) for debiasing and remove our assumption on test prior. Only given the zero-shot model and downstream training data without an assumption on test prior, **no algorithm**(including DebiasedPL) can guarantee an optimal performance for testing: your can always construct a test distribution that significantly differs from the training distribution to undermine the model performance. Therefore, your requirement is not reasonable.
> > >
> > > Secondly, DebiasedPL is general but not optimal in the fine-tuning settings. It is unreasonable to ask our method to work on every case, as we focus on fine-tuning not semi-supervised learning.
> > >
> > > Thirdly, our algorithm can be easily extended to imbalanced test set. Suppose the log probability for imbalanced test set is $\pi_t$, the GLA model is modified to $f_{gla}=softmax(f_{zs}(x)+f_{ft}(x)-\pi_p - \pi_s + \pi_t)$. As for noisy label, we think it unfair to ask our method to deal with it, it is clearly out of the scope of our work.
> > >
> > > We look for your reply if you have further concerns.

---

### Official Review · Reviewer_iLax · 2023-07-05

**Soundness:** 2 fair
**Presentation:** 2 fair
**Contribution:** 1 poor
**Rating:** 7
**Confidence:** 2

**Summary:**

This paper proposes a distribution estimation method that estimates the distribution of pre-training data for large VLMs like CLIP. Combined with downstream data distributions, a logit adjustment is applied to the outputs of both the zero-shot model and the fine-tuned model. By using a simple ensemble of the zero-shot and fine-tuned models, improved performance is achieved on the reported benchmarks.

**Strengths:**

The proposed method introduced performance improvements over compared baselines on most of the reported benchmarks.

**Weaknesses:**

Here are some questions:
1) The claim "their predictions are complementary to those of fine-tuned models" in line 30 cannot be supported by the cited references. Neither [3] nor [14] can support fine-tuned and zero-shot models are complementary.
2) In addition, in Section 4.2, where the authors attempt to demonstrate the complementarity of fine-tuned models and zero-shot models, the reasoning provided is not persuasive. The authors argue that fine-tuned models are biased towards downstream distributions, while zero-shot models are robust to distribution shifts. However, it remains unclear why a distribution shift-robust model should be complementary to a model that has already been adapted to a shifted distribution. If the authors claim and can prove that zero-shot models can provide something that is missing in the fine-tuned model and cannot be provided by the shifted downstream dataset, it needs to be explicitly demonstrated and proven. Otherwise, this assumption lacks solidity, which consequently undermines the validity of all subsequent proofs.
3) The rationale behind the method used to estimate the pre-training data distribution is not clearly explained. The paper does not provide sufficient information on why the method is designed in this particular way, nor does it address whether there are any means to verify the effectiveness of this method in accurately estimating the pre-training data distribution. Even though pre-training data is often proprietary, the authors could still perform toy experiments to demonstrate the efficacy of the proposed method. Given that this is a crucial component of the proposed method, it should not be overlooked.
4) The biggest weakness of this paper is the improvements introduced by the proposed method is very limited (~1% on average across all reported benchmarks).

Edit: most of my concerns are clarified. I will change my rating.

**Questions:**

Same as above.

**Limitations:**

The biggest limitation is the limited performance improvements.

---

> ### Author Rebuttal · Authors · 2023-08-09
>
> **Q1**The biggest weakness of this paper is the improvements are very limited.
> \end{point}
>
> **A1**We hold a different view. For ImageNet-1K full training with 1,000 classes, improving $1.5 \%$ accuracy over end-to-end fine-tuning is a noteworthy improvement. For ImageNet with 16 training shots, we observed a $3.5\%, 2.6\%, 2.1\%$ improvement over the previous prompt-tuning SOTA: CoOp, PLOT and ProGrad. On Place365-LT datasets, our GLA that only tuning prompt can even surpass the previous end-to-end fine-tuning SOTA BALLAD by a large margin ($47.2\%$ vs. $45.7\%$).
>     In light of these results, it is clear that our GLA demonstrates substantial gains across various datasets.
>
> ---
> **Q2**The claim "their predictions are complementary to those of fine-tuned models" in line 30 cannot be supported by the cited references. Neither [3] nor [14] can support fine-tuned and zero-shot models that are complementary.
>
> **A2** The underlying assumption of ensembling methods is that the individual models make independent errors[3,14] (See details in Section V.4 Independent Errors of [14]) otherwise the performance of ensembling will degenerate to the one of single model~(Imagine if we ensemble two identical models, the ensemble's performance will not improve). In the context of fine-tuning zero-shot models, this assumption is supported by empirical studies conducted in Section 5.1 of [1] which explores a series of measures of diversity, finding the complementarity of predictions between zero-shot and fine-tuned models.
>
> [1] Wortsman et al. Robust fine-tuning of zero-shot models. CVPR. 2022.
>
> ---
> **Q3** In addition, in Section 4.2, where the authors attempt to demonstrate the complementarity of fine-tuned models and zero-shot models, the reasoning provided is not persuasive. The authors argue that fine-tuned models are biased towards downstream distributions, while zero-shot models are robust to distribution shifts. However, it remains unclear why a distribution shift-robust model should be complementary to a model that has already been adapted to a shifted distribution. If the authors claim and can prove that zero-shot models can provide something that is missing in the fine-tuned model and cannot be provided by the shifted downstream dataset, it needs to be explicitly demonstrated and proven.
>
> **A3** We gracefully disagree. Assumption 1, diversity of zero-shot and fine-tuned models, often featured in theoretical analyses in distribution shifts [1-4] and empirically verified by prior work[5], appears to be a judicious assumption to adopt.
>
> Intuitively, zero-shot models and fine-tuned models leverage different cues to predict. For instance, zero-shot models rely on robust features for decisions that can achieve high performance on sketch and adversarial samples, while the fine-tuned models that trained on real images typically fail on these samples, as they rely on spurious correlations that only hold on real images.
>
> In fact, Assumption 1 is a weaker assumption compared to commonly used assumptions in  distribution shifts community [1,2,3]. In these papers, the in-distribution and out-of-distribution features are assumed to be disjoint parts of the inputs, each generated independently based on the label. In our paper, we only require the conditional independence of the outputs. In addition, we can find the same assumption to describe the relation between the fine-tuned models and the zero-shot CLIP in [5].
>
> Crucially, empirical studies detailed in Section 5.1 and Appendix Section E of [5] explore a range of diversity measures across predictions and features, revealing that zero-shot and fine-tuned models, despite a shared backbone, yield diverse predictions, which supports our claim.
>
> To sum up, we just follow the commonly used assumption in theoretical analysis of the robustness/out-of-domain generalization community, which is also empirically proven to be reasonable.
>
>     [1] Vaishnavh Nagarajan, Anders Andreassen, and Behnam Neyshabur. Understanding the failure modes of out-of-distribution generalization. ICLR, 2021
>
>     [2] Yining Chen, Colin Wei, Ananya Kumar, and Tengyu Ma. Self-training avoids using spurious features under domain shift. NeurIPS, 2020.
>
>     [3] Shiori Sagawa, Aditi Raghunathan, Pang Wei Koh, and Percy Liang. An investigation of why overparameterization exacerbates spurious correlations. ICML, 2020
>
>     [4] Ananya Kumar, Tengyu Ma, Percy Liang, and Aditi Raghunathan. Calibrated ensembles can mitigate accuracy tradeoffs under distribution shift. UAI, 2022.
>
>     [5] Wortsman et al. Robust fine-tuning of zero-shot models. CVPR. 2022.
> ---
>
> **Q4** The rationale behind the method used to estimate the pre-training data distribution is not clearly explained.
>
> **A4** The rationale behind the method (Step 1) to estimate pre-training label distribution has been justified in Proposition 2: the pre-trained label prior $\pi_p$ can be achieved when $f_{zs}-\log \mathbf{q}$ arrives at the lowest error $R_t$, as we prove that such model corresponds to the Bayes optimal classifier given fixed $f_{zs}$.
>
> ---
> **Q5** Even though pre-training data is often proprietary, the authors could still perform toy experiments to demonstrate the efficacy of the proposed method.
>
> **A5** To address your concerns, we devise a toy experiment where we have the true label bias at hand. In specific, we first train model on an imbalanced training set: CIFA10-LT~[1], then only use the testing set to estimate the label distribution. Figure 1 in the attached document illustrates a strong correlation between the estimated and real distributions, further supported by a small KL-divergence (0.00062). The toy experiments demonstrate the correctness of our proposed debiasing method.
>
> [1] Cui et al. "Class-balanced loss based on  effective number of samples." CVPR 2019.

---

> > ### Comment · Reviewer_iLax · 2023-08-21
> >
> > Thanks for the clarification. I was wrong at the performance. So I will change the final rating. The clarification for Q2,3 needs to be added into the main text.

---

> > > ### Author Response · Authors · 2023-08-22
> > > **Response to Reviewer iLax**
> > >
> > > We are grateful for your feedback and we will include the clarification for Q2 and Q3 to enhance the quality of our work.

---

### Official Review · Reviewer_8Trg · 2023-07-10

**Soundness:** 3 good
**Presentation:** 3 good
**Contribution:** 2 fair
**Rating:** 4
**Confidence:** 4

**Summary:**

Logit adjustment method has been widely used in long-tailed recognition, in which the source dataset is available, thus the class margin can be set in advance. However, the large-scaled foundational data cannot be accessed. To reduce the recognition bias of foundational models on downstream tasks, this paper proposes a generalized logit adjustment (GLA) method. The main contribution of GLA is to estimate the class margins for zero-shot models, after that, the fine-tuned and debiased zero-shot models are assembled together to achieve consistent improvement. The authors demonstrate the effectiveness of the proposed method through extensive experiments, including three tasks and three finetuning approaches.




**Strengths:**

1) Estimate the class margins for foundational models is an important task, especially when the pretrained datasets are not available. This paper proposes an effective generalized logit adjustment method for the zero-shot models.

2) To estimate the class margins, the authors consider it as a simple constrained optimization problem. They also provide theoretical analysis for their GLA method. The proofs seem to be ok.

3) The authors verify their method with extensive experimental results, including many-shot learning, few shot learning and long-tailed recognition tasks. The presented results deem to be good.

**Weaknesses:**

The main weaknesses include the following three aspects.

1) The proposed method is simple, and the theoretical proofs look a little reluctant.

a. Section 4.1 sees to be redundant. The Definitions and Lemma are less important for the following analysis.

b. The proof and conclusion of Proposition1 are not convincing.

c. There are some false assumptions about the discussion of Corollary 1. In case 1, the authors claim that the zero-shot model can not provide further improvement when the fine-tuned model is given. Obversely, this is wrong. Because their combination can further improve recognition accuracy.

2) The authors did not show the algorithm steps when estimating class margins (Eq. 3). They also did not provide the complexity and convergence of this optimization problem, and the choice of hyper-parameter \lambda.

3) Unfair comparisons make results look good. For example, in long-tail learning, the proposed GLA is an ensemble model with larger network parameters. Thus, it is not surprising that it outperforms other baselines. Please provide the model parameters and inference time to show the improvement.

Other comments:

1) Typo in line 109: w_k is not model parameter.

2) In Eq.4, v should be bold.


**Questions:**

1) How to initialize q?

2) Your assumption in Lemma 2 does not always hold, the target distribution may be unbalanced.

3) Please provide more analysis for your proofs. The constant C_2 in Eq. 21 and the conclusion of Eq. 30 are not easy to understand.

**Limitations:**

The authors discuss the limitations of their method.

---

> ### Author Rebuttal · Authors · 2023-08-09
>
> **Q1** The method is simple, and the proofs look a little reluctant.
>
> **A1** We hold a different view. **A simple method with statistical  grounding is our strength rather than weakness**. As Reviewer *wcSK* claimed ``The proposed method is both simple to understand and implement. This means that it could have a significant impact. Simple methods can easily be deployed by both practitioners and by researchers (as a strong baseline for future work). ''. In fact,  it is true evidenced by our experiment: despite simple, our method can easily beat more complicated methods like BALLAD, PLOT and ProGrad, which shows the value of our work. Regrading to proofs, we believe there exists some misunderstanding, and we elaborate the details as below to address your concerns.
>
> ---
> **Q2** Section 4.1 is redundant.
>
> **A2**  We gracefully disagree. Definition 1 and Lemma 1 are crucial. In Proposition 1, we first use Definition 1 to prove our GLA model is Bayes optimal. Then, we leverage Lemma 1 to prove our GLA model has the lowest risk. We use Definition 1 and Lemma 2 to prove Proposition 2. Without Section 4.1, the proof will be incomplete.
>
> ---
> **Q3** Explain the conclusion of Eq. 30
>
> **A3** First, we'd like to clarify that the conclusion of Eq.30 hinges on the preliminaries in Section 4.1. Therefore, the weakness  you raised, ``Section 4.1 seems to be redundant. '' stems from a misunderstanding of the  dependence between Section 4.1 and the subsequent proofs. Eq. 30 is restated:
>
> $\arg\max_{y}(f_{zs}(\mathbf{x})-\pi_p)_y $
>
> $= \arg\max_{y} softmax(f_{zs}(\mathbf{x})-\pi_p)_y$
>
> $=\arg\max_{y} P_t(y|f_{zs}(\mathbf{x}))$
>
> The first equality applies because $\text{softmax}(\cdot)$ does not change $\arg\max$ results. The second equality holds because of Eq.29. Recall that **Definition 3**: The Bayes Optimal classifier $y^*$ for $P$ given input $\mathbf{x}$  is defined as:
> $y^*(\mathbf{x}) = \arg\max_{y} P(y|\mathbf{x})$
>
> Given $f_\texttt{zs}(\mathbf{x})$ as the input, $\arg\max_{y}(f_\texttt{zs}(\mathbf{x}) - \pi_p)(y)$ is the Bayes optimal classifier, as it equals to $\arg\max_{y\in \mathcal{Y}}P_t(y|f_\texttt{zs}(\mathbf{x}))$.
> According to **Lemma 1**: The Bayes optimal classifier $y^*$ for $P$ has lower risk than all classifier $\hat{y}: \mathcal{X} \rightarrow \mathcal{Y}$. $\mathcal{R}(y^*)\leq \mathcal{R}(\hat{y})$.  Therefore we have the conclusion in Line 338-339: ``any other classifier $f_h(\mathbf{x})$ has higher risk than $f_\texttt{zs}(\mathbf{x})-\pi_p$.''
>
> ---
> **Q4** Explain C_2 in Eq.21
>
> **A4** Because $P_s(y|\mathbf{e})=\text{softmax}(\mathbf{e})_y$ and $P_s(y|\mathbf{z})=\text{softmax}(\mathbf{z})_y$. For some constants $C_s, C_p$, we can express $P_s(y|\mathbf{e})=\exp(\mathbf{e})_y/C_s$ and $P_p(y|\mathbf{e})=\exp(\mathbf{e})_y/C_p$. Replace this into Eq. (20), we get Eq. (21):
>
> $=\exp(\mathbf{e}+\mathbf{z}-\pi_s-\pi_p+\pi_t)_yC_1/(C_s \cdot C_p)=\exp(\mathbf{e}+\mathbf{z}-\pi_s-\pi_p+\pi_t)_yC_1/C_2$
>
> We denote $C_2=C_s \cdot C_p$ to simplify Eq. 21.
>
> ---
> **Q5** Target distribution may be unbalanced in Lemma 2
>
> **A5** We argue that it is very easy to extend our Lemma 2 to unbalanced target distribution.
>     In Eq.20 we incorporate $P_t(y)$ into constant $C_1$ as we assume $P_t(y)=1/K$ is constant. For an unbalanced target distribution, we modify Eq.20 into
>
> $P_t(y|\mathbf{e},\mathbf{z})=\frac{P_s(y|\mathbf{e})}{P_s(y)}
>         \frac{P_p(y|\mathbf{z})}{P_p(y)}P_t(y) C_1, $
>
> Let $\log P_t(y)=\pi_t(y)$, we rewrite Eq.21 as:
> $=\exp(\mathbf{e}+\mathbf{z}-\pi_s-\pi_p+\pi_t)_yC_1/C_2 $
>
> Using Eq.21-24, we arrive at our Lemma 2 under imbalanced target distribution:
> $ P_t(y|f_{ft}(\mathbf{x}),f_{zs}(\mathbf{x}))=softmax(f_{ft}(\mathbf{x})+f_{zs}(\mathbf{x}) -\pi_s-\pi_p+\pi_t)_y $
>
> ---
> **Q6**  How to initialize q? Algorithm steps in Eq. 3?
>
> **A6** We have detailed these in Line 383 and 384: q is initialized as a uniform distribution and optimized for 2,000 steps.
>
> ---
> **Q7** Complexity and convergence of the optimization.
>
> **A7** From Proposition 2, we know that $\mathbf{q}=\exp(\pi)$ is the minimizer of Eq.3. Thus, the optimization problem is guaranteed to have a solution. We use gradient descent to solve the problem, whose complexity is $O(nkm)$ where $k$ is the iteration steps, $n$ is the  sample size and $m$ is the label size.
>
> ---
> **Q8** hyperparameter lambda.
>
> **A8** We'd like to like to clarify that lambda is **not a hyperparameter**. lambda is updated to maximize the Lagrangian function in Eq.4 rather than a pre-defined hyper-parameter.
>
> ---
> **Q9**  In case 1, the authors claim that the zero-shot model can not provide further information when the fine-tuned model is given. Obviously, this is wrong.
>
> **A9** There seems to be a misunderstanding. In fact, **we are on the same page: we also claimed Case 1 is unlikely to happen**. Case 1 discusses the situation when $R_t(f_{gla})=R_t(f_{ft})$, i.e., our GLA models degenerate to fine-tuning models. This situation happens when ``the zero-shot model can not provide further improvement when the fine-tuned model is given''. As you claims, this is not likely to happen, which supports our claims.
>
> ---
> **Q10** Ensemble is unfair.
>
> **A10** We see your point, but we gracefully disagree. First, our GLA does not involve external data or models, all baselines are given the same zero-shot models and fine-tuning data. Second, our baselines include other ensemble methods, e.g. WiSE-FT and naive ensemble, and our GLA shows superiority. Given that we share the same model parameter and inference time as these ensemble baselines, we believe the comparison is fair.
>
> ---
> **Q11** w_k is not model parameter
>
> **A11** We're sorry that you misunderstood the point. w_k is the parameter of classification head initialized by prompting: it is updated during fine-tuning.
>
> ---
> **Q12**  v should be bold
>
> **A12** That is not true. We only have **one** equality constraint in Eq.3: $1-\sum_{i\in [K]} \textbf{q}_i$. Therefore, v should be a scalar rather than be bold.

---

### Official Review · Reviewer_wcSK · 2023-07-26

**Soundness:** 3 good
**Presentation:** 3 good
**Contribution:** 3 good
**Rating:** 7
**Confidence:** 3

**Summary:**

This paper proposes a method for combining the predictions of zero-shot and few-shot classifiers. The method removes the need for a weighting hyper-parameter to interpolate between the predictions. Furthermore, it also removes a source of bias (due to the frequency of words in the pre-training dataset).

**Strengths:**

## Simplicity

The proposed method is both simple to understand and implement. This means that it could have a significant impact. Simple methods can easily be deployed by both practitioners and by researchers (as a strong baseline for future work).

## Presentation

The presentation of the method is clear, concise, and easy to understand.

## Experimental Results

The experimental results show that the proposed method improves over a selection of sensible baselines in a range of settings, with some strong improvements in several cases.


**Weaknesses:**

## Correctness

While the proposed method clearly results in improved performance in several settings, it is not clear to me that the reason for the improvement suggested by the authors is correct. That is, I am not convinced that the proposed method to estimate the label bias of the pre-training dataset is working as intended. Does $\log \mathbf{q}$, αs described in eq 3 and sec 4.3, provide a good estimate of $\pi_p$? I do not believe that the paper provides theoretical or empirical evidence that this is the case and that the mechanism behind the success of the method is due to the suggested bias improvement, rather than something else.

##############

### Edit, after rebuttal and discussion.

The authors have now addressed my concerns above. They have demonstrated that their approximation is accurate when the downstream and pre-training datasets are similar. They have also shown that when the downstream and pre-training datasets are different, the approximation becomes worse. Nonetheless, the proposed method still works well, and the authors have given intuition for why this might be the case. Given the authors proposed updates to the paper, I believe that the story of the paper will now be accurate and clear. I've raised my score from 5 to 7.

**Questions:**

Additional experimental and theoretical analysis showing that $\log \mathbf{q} \approx \pi_p$ would significantly strengthen the paper. Concretely, if this weakness were addressed, I would happily increase my score by 1-2 points.

The true value of $\pi_p$ could, for example, be estimated by using the LAION-400m dataset, which is similar in scale and content to the true pre-training dataset of CLIP and results in similar zero-shot accuracies. This was the approach taken in the "A Simple Zero-shot Prompt Weighting Technique to Improve Prompt Ensembling in Text-Image Models" paper (https://arxiv.org/abs/2302.06235), which addresses similar biases in the context of prompt selection in zero-shot classifiers.

Additionally, including the above-mentioned paper in the related work would be useful.





**Limitations:**

The theoretical and experimental analysis does not support the main claim of the paper, as discussed above under "weaknesses".

---

> ### Author Rebuttal · Authors · 2023-08-09
>
> **Q1** Additional experimental and theoretical analysis showing that $\log \mathbf{q}=\pi_p$
>
> **A1**
> To further address the concerns about our bias estimation, we further provide explanation of our Proposition 2 and some empirical evidences.
> In Proposition 2, we proved that when given the outputs of the zero-shot model $f_\texttt{zs}(\mathbf{x})$, $f_\texttt{zs}(\mathbf{x}) - \pi_p$ will be Bayes optimal classifier for the downstream distribution $P_t$, therefore it is the minimizer of $R_t(f_h(\mathbf{x}))$. In Proposition 2, $f_h(\mathbf{x})$ is defined as the arbitrary classifier that uses $f_\texttt{zs}(\mathbf{x})$,  it includes the hypothesis of $f_\texttt{zs} - \log \mathbf{q}$. Therefore, $\pi_p$ is also the minimizer of Eq.(3). By solving the constrained optimization problem, we can get a good estimation of pre-training label bias.
> We subsequently provide two pieces of evidence to further validate the correctness of label bias estimation.
>
> **Evidence 1: Experiments to compare the true label bias and the estimated one.**
> We devise more intuitive experiments where we have the true label bias at hand. In specific, we first train models using ResNet32 backbones on an imbalanced training set: CIFAR10-LT [1], then only use the test set to estimate the label distribution.  Figure 1 in the attached document illustrates a strong correlation between the estimated and real distributions, further supported by a small KL-divergence (0.00062). The experiments demonstrate the correctness of our proposed debiasing method.
>
> **Evidence 2: Estimated $\pi_p$ is transferable across different zero-shot models.** In Section 5.1 and C.1, we demonstrate that when different models are pre-trained on the same pre-training dataset, the pre-trained bias estimated by one zero-shot model (e.g. CLIP-ViT-B/32) can be subsequently transfered to debias other models (e.g. CLIP-ViT-B/16 and CLIP-ViT-B/14). As shown in Table 9, debiased CLIP-ViT-B/{16, 14} show clear performance gains over original zero-shot models.
>
> ---
>
> **Q2**  Discuss [2] in the related work.
>
> **A2**    We thank the reviewer for the suggestion.
>     [2] automates the prompt engineering by prompt scoring, which also targets on alleviating word frequency bias in pre-training data. Our approach diverges from [2] in two main aspects.
>     The first difference is that our approach focuses on debiasing zero-shot models given fixed prompts, contrasting with [2] that optimizes prompting process. Secondly, unlike [2] that necessitate access to a subset of the pre-training data, our GLA is exempt from this requirement. We will include the discussion in the related work in the revised version.
>
> ----
>
> [1] Cui et al. "Class-balanced loss based on effective number of samples." CVPR 2019.
>
> [2] Allingham et al. A Simple Zero-shot Prompt Weighting Technique to Improve Prompt Ensembling in Text-Image Models

---

> > ### Comment · Reviewer_wcSK · 2023-08-10
> >
> > Thanks for your response. Unfortunately, I am still unconvinced. To be clear, I am not arguing with the point that $f_{zs}(\mathbf{x}) - \pi_p$ is the Bayes optimal classifier, nor am I arguing against $f_{zs}(\mathbf{x}) - \log\mathbf{q}$ being included in $f_h(\mathbf{x})$. What is not clear to me is that $\pi_p$ is approximated by $\log \mathbf{q}^*$. Proposition 2 simply tells us that the risks should be the same, on average. It is not apparent to me that this means that $\pi_p = \log \mathbf{q}^*$. On a more conceptual level, it would be very surprising to me that you can estimate the marginal log-probs of the pre-training distribution without access to pre-training data. Thus, I hypothesize that your method's good performance is not due to estimating $\pi_p$ well but due to some other mechanism.
> >
> > While Evidence 1 does somewhat support your claim, it is not convincing enough. This experiment is very different from the setting of interest. In particular, you are using training and test inputs from the same distribution, even if the labels have different distributions. It is unclear whether you would see the same behavior for a zero-shot classifier trained on a large and broad "internet scale" dataset.
> >
> > On the other hand, I don't find Evidence 2 compelling. I think these results highlight a definite strength of your method. Still, they don't necessarily suggest that $\pi_p$ is being estimated, just that the quantity $\log \mathbf{q}^*$ is transferable, which could be due to any number of other reasons.
> >
> > As I mentioned in my review, an experiment that I would find compelling is to estimate $\pi_p$ using the LAION dataset. Allingham et al. find that even using 20K images from LAION, the prompt selection for a zero-shot classifier can be debiased.
> >
> > There is something that I need to clarify, which may impact my thoughts above. How can we have a marginal distribution (i.e., $P_p(y)$) over labels under the pre-training distribution? The pre-training dataset for CLIP doesn't have any notion of the classes from the training or test distributions. Assuming we had access to the pre-training dataset, would we estimate $\log P_p(y)$ by averaging the logits of the constructed zero-shot classifier over all of the images in the training data?
> >
> > I look forward to your response.

---

> > > ### Author Response · Authors · 2023-08-11
> > > **Response to wcSK**
> > >
> > > Thanks for your prompt and valuable response.
> > >
> > > A. Further explanation on $\pi_p$:
> > > We agree with your intuition.
> > > The $\log\mathbf{q}$ we estimated is not the marginal log-probability over the entire pre-training distribution but the label bias aligns with the downstream distribution.
> > > In Evidence 1, as you pointed out, while the training and test sets have different label distributions, their conditional distribution $P(x|y)$ remains invariant. In such situation, our estimation is guaranteed to converge to the true training label bias.
> > > For CLIP models, the pre-training data are diverse. It is likely that some of the pre-training data fall out of the distribution of downstream domain and leads to an inaccurate estimation for entire pre-training distribution.
> > >
> > > However, we'd like to point out that removing the label bias of entire pre-training distribution is not optimal for downstream tasks. For instance, suppose the pre-training dataset contains "sketch'' and "photo'' styles for "dog'' and "cat'' samples. Suppose the sample size of "dog'' and "cat'' is equal but there are much more "sketch dogs'' that "sketch cats''. In other words,  although the entire distribution is class-balanced,  each style domain is imbalanced, leading to biased zero-shot predictions within each domain. In such scenario, it we want to deploy the zero-shot models in "sketch dogs and cats'' domain, removing the balanced label bias of entire pre-training distribution is ineffective. The optimal label bias should be estimated on the ``sketch'' distribution.
> > >
> > >
> > > B. Experiment using LION-400: We are currently estimating using LION-400. Once the experiments are complete, we will append the results.
> > >
> > > C. How to estimate $P(y)$ when we have the access to the pre-training data: We believe that averaging the logits across all training images provides a good estimation of $\log P(x)$. Although CLIP does not has a notion classes, it can be prompted with a class name to approximate $P(y|x)$. Therefore, average outputs across all training images is doing $E_{x \sim P_p}[P(y|x)]=P_p(y)$.

---

> > > > ### Author Response · Authors · 2023-08-16
> > > > **Response to wcSK**
> > > >
> > > > Dear Reviewer,
> > > >
> > > > Following [2], we retrieved the first 20k images and used the open clip ViT-B/16 to estimate the expected logits across 8 classes: dog, cat, squirrel, tiger, elephant, horse, pig and bird. We adopted the prompt "a photo of a \{\}".
> > > > For our label bias estimation, we consider two image styles "photo'' and "sketch'' from DomainNet[3] dataset. For each domain, we randomly sampled 50 images for each class.
> > > >
> > > > We present the expected logits estimated by [2], along with the $\pi_p$ calculated from ''photo'' and ''sketch'' downstream domain data in the following table.
> > > > Since the softmax is invariant to constant offsets, i.e. $\text{softmax}(x+c)=\text{softmax}(x)$, we align the three label bias to yield the same logits on ``dog'' class by subtracting constant values.
> > > >
> > > > |                        | dog   | cat   | squirrel | tiger | elephant | horse | pig   | bird  |
> > > > |------------------------|-------|-------|----------|-------|----------|-------|-------|-------|
> > > > | expected logits by [2] | 0.059 | 0.039 | 0.043    | 0.043 | 0.043    | 0.062 | 0.061 | 0.028 |
> > > > | $\pi_p$ by ``photo"    | 0.059 | 0.041 | 0.047    | 0.055 | 0.048    | 0.062 | 0.060 | 0.033 |
> > > > | $\pi_p$ by ``sketch"   | 0.059 | 0.043 | 0.063    | 0.061 | 0.067    | 0.042 | 0.047 | 0.056 |
> > > >
> > > > We observe that the label biases estimated by the "photo" domain align closely with [2] due to its close resemblance to the pre-trained domain.
> > > > Conversely, the "sketch" image style, which significantly differs from the pre-training domain, results in a more pronounced deviation in the label biases estimated by [2].
> > > >
> > > > Additionally, we applied method [2] and our GLA to debias the zero-shot model. The results are shown in the following table, where the superiority of our method becomes evident (93.00\% vs $92.25\%$). Applying the label bias from [2] on the "sketch" domain degrades the model's performance (from 92.25\% to 89.50\%). This is attributed to the overall pre-training label bias does not adequately reflect the bias specific to the "sketch" domain.
> > > >
> > > > |                     | Accuracy |
> > > > |---------------------|----------|
> > > > | zero-shot openclip  | 92.25    |
> > > > | debiased by [2] | 89.50    |
> > > > | debiased by GLA     | 93.00    |
> > > > *Accuracy on Sketch Domain*
> > > >
> > > > [3] Peng et al. Moment Matching for Multi-Source Domain Adaptation. ICCV 2019.

---

> > > > > ### Comment · Reviewer_wcSK · 2023-08-16
> > > > >
> > > > > Thank you for the updates and the new experimental results. I think these discussions and results have been very fruitful and have given me a much better understanding of your method. As a result, I am strongly considering increasing my score for the paper. My only remaining concern is that these new results and perspectives must be inserted into the narrative correctly to give readers of the paper the complete picture.
> > > > >
> > > > > In my opinion, there are two main changes required for the paper:
> > > > > 1. Any discussion with regards to the estimation of $\pi_p$ must be subtlety adjusted to make the following points:
> > > > >     * $\log \mathbf{q}$ only approximates $\pi_p$ accurately when the downstream data is similar to the pre-training dataset (i.e., the images come from similar domains, such as both being photographs rather than one being photos and the other sketches).
> > > > >     * Approximating $\pi_p$ might not actually be desired (i.e., it might not provide the best performance), e.g., when the downstream data does not match the pre-training data.
> > > > > 2. The LAION experiment above (and, optionally, some of the other additional evidence you've provided) should be included in the main text to support the ideas above.
> > > > >
> > > > > Do you agree with this suggestion at a high level? And if so, do you have any concrete descriptions of what you plan to change and where?
> > > > >
> > > > > By the way, I (and I think the future readers of the paper) would be interested to see the accuracies on the photo domain to compare with the sketch domain results above.

---

> > > > > > ### Author Response · Authors · 2023-08-17
> > > > > > **Response to Reviewer wcSK**
> > > > > >
> > > > > > We are grateful for your feedback and we agree with your suggestions to enhance the quality of our work.
> > > > > >
> > > > > > In response to the two changes:
> > > > > >
> > > > > > 1. We plan to append the discussion in Section 4.3. We will first point out the situation when our estimated $\log q$ accurately approximates $\pi_p$, that is the downstream data is similar to the pre-training one. Afterwards, we will admit that when the above situation violates, the estimation for $\pi_p$ will be inaccurate. In the end, we will indicate that approximating $\pi_p$ might not actually be desired for downstream tasks. When addressing downstream tasks, our focus is the label bias of the pre-trained model specifically on data relevant to the downstream distribution, not the whole distribution.
> > > > > >
> > > > > > 2. We intend to insert a new section after Section 4.3 to include the CIFAR-10-LT and the LAION experiments. We plan to showcase the ``ideal'' situation -- label bias estimation for CIFAR-10-LT, where the images come from similar domain. In this context, our $\log q$ accurately approximates the true $\pi_p$. Subsequently, we will provide the experiments on the LAION dataset, illustrating the situation when the downstream data diverges from the pre-training set. In such situation, we point out that approximating $\pi_p$ might not be optimal for downstream performance, but our estimation of $\log q$ is desirable.
> > > > > >
> > > > > > ---
> > > > > >
> > > > > > The accuracies on photo domain are presented in the following table.
> > > > > >
> > > > > > |Method|Acuraccy|
> > > > > > |---|---|
> > > > > > |zero-shot openclip| 97.00 |
> > > > > > |debiased by [2] | 97.50|
> > > > > > |debiased by GLA | 97.75 |
> > > > > > Accuracy on Real Photo Domain

---

> > > > > > > ### Comment · Reviewer_wcSK · 2023-08-17
> > > > > > >
> > > > > > > That sounds good to me – I am happy to update my score. Thanks for your engagement with my review.
> > > > > > >
> > > > > > > (The photo domain results complement the sketch results and your conclusions nicely).

---

### Author Rebuttal · Authors · 2023-08-09

# Response to All Reviewers

Dear Program Chair, Senior Area Chair, Area Chair, and Reviewers,

First of all, we gratefully thank all the reviewers for their thoughtful comments and feedback.

In this paper, we identify label bias in foundation models like CLIP and underscore its adverse effects on downstream task performance. We propose the Generalized Logit Adjustment (GLA) framework for fine-tuning foundation models, which boosts the performance by effectively eliminating label bias. The contribution of this paper is four-fold:

**1. Solid Theoretical Analysis:** We formalize the estimation of label bias as a constrained optimization problem and prove that our GLA model is a Bayes optimal classifier (Proposition 1).

**2. Simple and Practical Solution:** Our GLA is a post-hoc approach without introducing hyper-parameters, which makes it  easy to implement. In addition, we does not necessitate access to the pre-training dataset, which makes it practical for fine-tuning scenarios.

**3. Comprehensive Evaluation:** We consider three real-world settings and three fine-tuning scenarios, conducting experiments on over 20 datasets.

**4. Significant Performance Improvement:**  The GLA method demonstrates significant improvements, e.g., it achieves 1.5 pp accuracy gains on ImageNet and large average improvement (1.4-4.6 pp) on 11 few-shot datasets.

We add a PDF that contains additional experimental analysis on bias estimation: we devise an experiment to demonstrate the estimated label distribution strongly approximates to the true one.

As our paper received mixed ratings, i.e., three positive (755) and two negative (44), it would be appreciated if the reviewers could have a look at our responses and revision. We have tried our best to address your concerns in our responses in detail. Hope that our responses answered the questions. Please let us know at your early convenience if you have further questions or concerns.

Best regards,

Authors of Paper \#6116

---

### Decision · Program_Chairs · 2023-09-21

**Decision:**

Accept (poster)

**Comment:**

There was a significant split between the reviewers—three were very positive about the work, while the other two had significant concerns. However, one of these negative reviewers did not engage with the authors' rebuttal or with the other reviewers, and during the final discussion phase two of the more positive reviewers argued convincingly (and I agree) that those concerns were largely addressed by the authors' rebuttal. Overall, this seems like a solid contribution that deserves to be accepted.